# A sum of its parts: A systematic review evaluating biopsychosocial and behavioral determinants of perinatal depression

Kayla D. Longoria[1,2]*, Tien C. Nguyen[3,4], Oscar Franco-Rocha[1], Sarina R. Garcia[3], Kimberly A. Lewis[1,2], Sreya Gandra[3,5], Frances Cates[5], Michelle L. Wright[1,6]

1 School of Nursing, University of Texas at Austin, Austin, Texas, United States of America, 2 Department of Physiological Nursing, School of Nursing, University of California, San Francisco, San Francisco, CA, United States of America, 3 College of Natural Sciences, University of Texas at Austin, Austin, Texas, United States of America, 4 University of Pittsburgh School of Medicine, Pittsburgh, Pennsylvania, United States of America, 5 College of Liberal Arts, University of Texas at Austin, Austin, Texas, United States of America, 6 Department of Women's Health, Dell Medical School at The University of Texas at Austin, Austin, Texas, United States of America

* kayla.longoria@ucsf.edu

**Data Availability Statement:** All relevant data are within the manuscript and its Supporting Information files.

## Abstract

### Introduction

Depression is one of the most common yet underdiagnosed perinatal complications and our understanding of its pathophysiology remains limited. Though perinatal depression is considered to have a multifactorial etiology, integrative approaches to investigation are minimal. This review takes an integrative approach to systematically evaluate determinants (e.g., biological, behavioral, environmental, social) and interactions among determinants of perinatal depression and the quality of methods applied.

### Methods

Four databases (i.e., PubMed, CINAHL, APA PsycInfo, Web of Science) were systematically searched to identify studies examining determinants of perinatal depression in adult perinatal persons ($\geq$ 18 years). Articles were excluded if the outcomes were not focused on perinatal persons and depression or depression symptoms, depression was examined in a specific subpopulation evidenced to have psychological consequences due to situational stressors (e.g., fetal/infant loss, neonatal intensive care unit admission), or was considered grey literature. The Critical Appraisal Skills Programme and AXIS tools were used to guide and standardize quality appraisal assessments and determine the level of risk of bias.

### Results

Of the 454 articles identified, 25 articles were included for final review. A total of 14 categories of determinants were investigated: biological (5), behavioral (4), social and environmental (5). Though only 32% of studies simultaneously considered determinants under more than one domain, a pattern of interactions with the tryptophan pathway emerged. Concerns

**Funding:** KDL was supported by the National Institute of Nursing Research of the National Institutes of Health (NIH) under Award No. T32NR019035. https://www.ninr.nih.gov/ The funders had no role in study design, data collection and analysis, decision to publish, or preparation of the manuscript.

**Competing interests:** The authors have declared that no competing interests exist.

for risk of bias were noted or were unclear for three types of bias: 13 (52%) selection bias, 3 (12%) recall bias, and 24 (96%) measurement bias.

## Conclusions

Future research is needed to explore interactions among determinants and the tryptophan pathway; to strengthen the methods applied to this area of inquiry; and to generate evidence for best practices in reporting, selecting, and applying methods for measuring determinants and perinatal depression.

## Introduction

The leading underlying cause of perinatal death is mental health conditions [1]. Depression is one of the most common conditions to occur perinatally as it impacts every one in five perinatal persons [2, 3]. Perinatal depression denotes the manifestation of affective, somatic, and/or cognitive symptoms, ranging in severity, that can occur at any time point in the perinatal period (i.e., conception-12 months postpartum) and impairs one's ability to complete daily activities [2–4]. While impairment in functioning is already of concern due to the increased physiological, psychological, and financial demands generated by this life-stage, distal outcomes (i.e., suicide, opioid use disorder) continue to contribute to the alarmingly unabated maternal mortality rates in the US where 80% of these deaths are considered preventable [1, 5, 6]. For instance, suicide, a leading cause of maternal mortality, has tripled over the last decade and accounts for ~20% of perinatal deaths [6, 7], whereas opioid use disorder accounts for one of the most frequent causes of accidental death [1, 5, 8]. Yet, depression remains the most underdiagnosed perinatal complication in the US [2] suggesting advancements in our understanding of the risk for and development of the condition requires timely attention and response.

The heterogeneous nature of depression symptoms coupled with the stark overlap of "normal" pregnancy symptoms make early detection and intervention difficult. Therefore, the prevalence of perinatal depression is likely underrepresented in part due to the lack of diagnostic expertise in the clinicians who are most likely to interact with at-risk individuals, high variability in existing screening practices, and underreporting of symptoms due to perceived stigma [9–11]. Still, 10–20% of perinatal persons are reported to experience depression [3, 12–14].

Siloed approaches to investigation may inadvertently omit significant findings related to interactions among factors that could provide a deeper understanding of disease risk and onset. In an era of team science, integrative approaches to investigation, though complex, are perceived as desireable to address some of the world's most complex health problems [15, 16]. Since perinatal depression is considered to have a multifactorial etiology, integrative approaches to investigation may be ideal to aid in bridging knowledge gaps and lead to advancements in detection and intervention. However, the few studies that have been conducted to investigate both biological and external factors that contribute to perinatal depression demonstrate integrative approaches to investigation have been minimal [13, 17].

Though evidence suggests interactions among external factors and biological factors can contribute to the onset of pathology [18], the factors most commonly explored in the etiology of perinatal depression have been external (e.g., social determinants of health, personal or family history of a psychiatric condition, low socioeconomic status (SES), stress, poor social support, intimate partner violence) [12, 13, 17, 19–23]. Due to the limited understanding of

biological factors that may contribute to depression in perinatal populations, biological theories of depression in the general population (i.e., immune response, inflammation, tryptophan metabolism) may be useful in informing initial directions for investigations including biological factors in perinatal specific depression [24–28].

To our knowledge, no prior reviews examining both biological and external determinants of perinatal depression have specifically included the tryptophan pathway. This review aggregates existing literature across various scientific domains and uncovers novel interactions among biological and external factors that warrant further investigation into the etiology and risk for this complex condition. Advancements in knowledge of distinct determinants and interactions will not only improve our ability to detect existing symptoms but will also progress our aptitude for determining risk status and implementing risk mitigation strategies [10]. Therefore, the purpose of this review is to take an integrative approach to systematically evaluate a) what social, environmental, behavioral, and biological determinants (i.e., immune response, inflammation, tryptophan metabolism) have demonstrated a relationship with perinatal depression b) how such determinants effect perinatal depression, and c) the quality of the methods used in the included studies.

## Methods

### Search strategy

The literature search took place in December 2022. The following databases were searched for articles that encompassed all or some of the specified determinants: PubMed, CINAHL, APA PsycInfo, and Web of Science. The following search terms were used across all databases in the Title/Abstract field: (depression or depressive or mdd or major depressive disorder or clinical depression or unipolar depression) AND (social or environmental or behavioral) AND (determinants or characteristics or factors) AND (tryptophan or serotonin or kynurenine or immunology or immune response or immune system or inflammation or inflammatory response or cytokines) AND (metabolites or metabolomics or metabolism) AND (pregnan* or prenatal or perinatal or antenatal or postpartum or postnatal or matern* or peripartum or intrapartum).

The study selection process was guided by the Preferred Reporting Items for Systematic Reviews and Meta-analyses (PRISMA) methodology [29]. Search results and duplicates were managed using the open-source reference management software Rayyan [30]. Microsoft Excel was used as a screening and data extraction tool to organize articles among the six authors (KDL, MLW, SG, TCN, KL, OFR), and allowed the primary author to successively cross-check articles screened to confirm eligibility decisions before proceeding to full-text review and quality appraisal.

### Inclusion and exclusion criteria

Articles from any date were included if they focused on a timeframe within the perinatal period (i.e., conception-12 months postpartum), had participants that were 18 years or older, were available in the English language, investigated factors that belonged to at least one of the four domains (i.e., biological, behavioral, environmental, social), and had an outcome of depression or depression symptoms. We define the four domains as follows: 1) biological: individual features unique to a person that have a biological basis (e.g., genetics, brain chemistry, hormone levels) 2) behavioral: either a conscious or unconscious action or inaction in response to internal or external stimuli (e.g., dietary intake, smoking, physical activity) 3) environmental: physical surroundings or conditions a person lives or functions within (e.g., access to resources, air pollution, poor water quality, crime) 4) social: one's experiences with

relationships or interactions with others (e.g., racism/discrimination, intimate partner violence, social support) [31, 32].

Articles were excluded if they were non-peer-reviewed publications, review/meta-analyses, and commentaries. Further, articles were excluded if outcomes were not specific to pregnant/postpartum individuals (i.e., partner, support persons, infant), determinants investigated were not related to depression or depression symptoms (e.g., post-traumatic stress disorder). We also excluded studies where depression or depression symptoms were examined in a specific subpopulation of participants experiencing situational stressors evidenced to have psychological consequences (e.g., fetal/infant loss, neonatal intensive care unit admission). We perceived including studies focused specifically on subpopulations experiencing situational stressors that are suggested to have psychological consequences to have potential to convolute the results [33, 34].

## Article selection and quality appraisal

After all articles were compiled and duplicates were removed, six authors (KDL, MLW, KAL, SG, TCN, OFR) independently screened the titles and abstracts to determine which articles met inclusion criteria. All articles were then subsequently cross-checked by the primary author to make a final determination on inclusion. Of the articles that remained after the title and abstract screening, five authors (KDL, MLW, SG, TCN, OFR) independently completed a full-text review. Any concerns related to inclusion during any of the screening processes were resolved by discussion among the primary author and the respective co-author.

Quality appraisal screening was independently conducted by two authors (KDL, TCN) to ascertain any methodological or risk of bias concerns. Since quality appraisal assessments can be subjective in nature, we selected two commonly used quality appraisal tools (i.e., Critical Appraisal Skills Programme [CASP] and AXIS), respective to study design, to guide and standardize the process [35, 36]. CASP was selected because the tool comes in the form of individual checklists that include questions tailored to specific study designs and are organized in a way that promotes efficiency in the appraisal screening process. This checklist was also considered desirable, compared to similar checklists (e.g., JBI), because each question has a comment section where screeners can document their thought process when answering each question [35–37]. Comments were thought to be useful in the case there were any discrepancies, that would warrant further discussion, between the two screeners in the screening process. The checklist was perceived as efficient since the questions are organized in sections that first allow users to quickly examine the validity of the study design and whether the study was methodologically sound before proceeding to assess any findings. Lastly, since CASP does not have a form for cross-sectional studies, we selected AXIS to not only filled that gap but also because it had a similar checklist structure which was desired for the aforesaid reasons [36]. Once the studies underwent quality appraisal screening, they were then categorized as having a low, moderate, high, or unclear risk of bias per three types of bias (i.e., selection bias, recall bias, measurement bias). The types of bias and levels of risk are defined in **Table 1**.

## Data extraction and synthesis

The following data were extracted from the included articles: country, purpose or aims, study design, recruitment and sampling method, perinatal period investigated, number of time points, sample description, what determinants were investigated, methods for measuring determinants and depression, method of analysis, and findings related to the relationship among determinants and depression or depression symptoms. Once data extraction was

**Table 1. Definitions of types of bias and level of risk.**

| Term | Definition |
|---|---|
| Selection bias | any non-random error in methodological decisions that influence how a study sample is acquired. |
| Recall bias | occurs when the data collected from the participant may not be an accurate representation of the event or information being investigated given the lapse in time from when the event occurred to when the participant is being asked to recount information about the event. |
| Measurement bias | any non-random error in how an outcome is measured or evaluated. |
| Low risk of bias | sufficient information about the methods of investigation is provided, and there are minimal concerns related to risk of bias that could compromise the validity of the findings. |
| Moderate risk of bias | a majority of information about the methods of investigation are provided and/or a few concerns related to risk of bias were noted that could potentially influence the validity of the findings. |
| High risk of bias | a significant amount of essential information about the methods of investigation are not provided and/or a considerable number of concerns related to risk of bias were noted that likely compromise the validity of the findings. |
| Unclear risk of bias | too few methodological details were reported by the investigators to allow for a genuine determination of the level of risk of bias. |

complete, the data were organized by descending date respective to the time-period investigated (i.e., pregnancy, postpartum, perinatal) and then synthesized.

## Results and discussion

The PRISMA flow diagram provides an overview of the search results (**Fig 1**). Twenty-six articles remained for full-text review and quality appraisal screening. One article was excluded [38] during quality appraisal screening due to methodological concerns making the total articles included 25 [39–63]. The article by Handley and colleagues (1980) was excluded for the following reasons: a) terms for perinatal timeframes were not clearly or consistently stated; b) affective symptoms were measured rather than depression; c) four different affect measures were used to determine cases versus non-cases; d) inconsistent measurement of cases (i.e., considered a case if they measured above $80^{th}$ percentile on any or all four affect measures); e) no rationale provided for cut off decisions or methods for determining cases; f) measures used were those that are not widely used in depression or perinatal depression research or practice; g) insufficient description of methods for measuring biochemicals; h) methods for measuring the primary outcome and biochemicals were from roughly half a century ago.

## Description of study and sample characteristics

Of the 25 articles included in the final review, 80% of the articles were published in 2011 or later [39, 41, 43–48, 50, 51, 54–63]. Though the US maternal mortality rates continue to markedly exceed that of other high-income countries [64], over half (60%) of the studies [40, 41, 43–45, 47, 49, 51, 52, 54, 55, 59–62] were conducted outside of the US. Overall, sample sizes ranged from 16 to 3,252 (N = 9,481). Notably, sample sizes were much lower in studies conducted in the US (n = 1,407, M = 141, SD = 127) [39, 42, 46, 48, 50, 53, 56–58, 63] compared to non-US based studies (n = 8,074, M = 538, SD = 872.4) [40, 41, 43–45, 47, 49, 51, 52, 54, 55, 59–62]. To determine if the difference was statistically significant, a Mann-Whitney U Test was performed using the open-source software tool R v.2022.12.0+353 but did not demonstrate a statistically significant difference (U = 50, p = 0.1775).

A total of 21 studies [39, 41–44, 46–63] reported sample age (88%), with the mean age of participants being 29.49 (2.71) years. Race and/or ethnicity was reported in nine [42, 46, 48, 50, 54, 56–58, 63] of the 25 studies (36%), but only five studies [46, 50, 56, 57, 63] (20%)

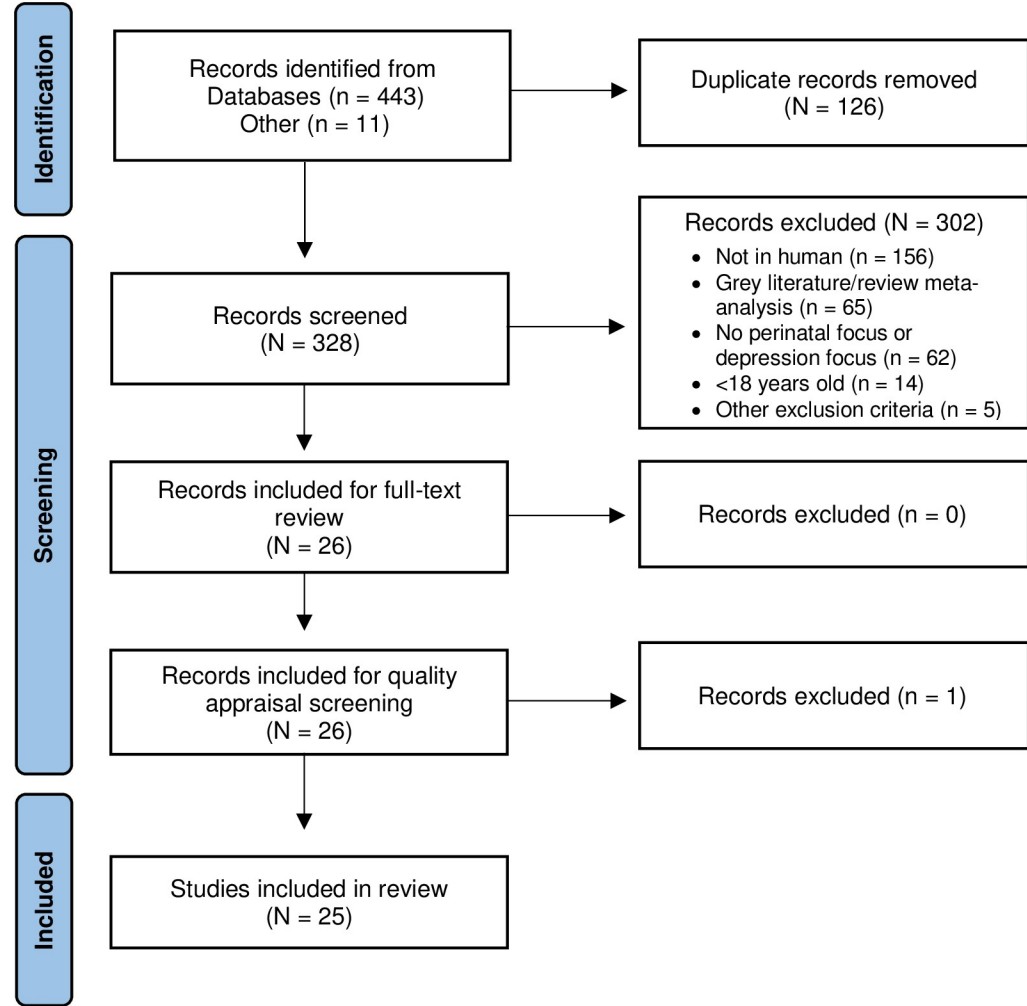

**Fig 1. PRISMA flow diagram.**

included race and/or ethnicity in the analyses. Further, eight [42, 46, 48, 50, 54, 56–58, 63] of the nine studies that reported race and/or ethnicity were studies conducted within the US and 66.7% of those studies [46, 48, 50, 57, 58, 63] had samples comprised predominantly of Non-Hispanic White individuals. Meaning, existing knowledge on determinants of perinatal depression may exclude minority populations which have the highest rates of perinatal depression and maternal mortality and morbidity in the US [1, 11, 64–66]. Of the 15 studies [41, 42, 46–48, 50, 51, 54–59, 61, 63] that reported participant education, 80% [41, 46, 48, 50, 51, 55–59, 61, 63] had samples primarily comprised of individuals with at least some college education. Nine studies [42, 44, 46, 47, 50, 51, 54, 55, 58] reported income and/or SES with 88.8% [42, 44, 46, 47, 50, 51, 55, 58] including a significant number (i.e., ≥50% of total sample) of participants from low to middle class. Nearly half (44%) of the studies [42, 44, 45, 51, 55–58, 61–63] reported parity with only one study [55] specifically looking at first-time mothers.

These demographic factors are important to consider because current evidence suggests those from lower SES and/or first-time mothers may be at increased risk of developing perinatal depression [13, 21, 23, 67]. However, there is conflicting evidence for level of education being a risk factor versus a protective factor [68]. Conflicting findings may be a result of how

education is operationalized and/or the heterogeneity among other sample characteristics (e.g., race and ethnicity, country) and the condition itself. Demographic information is collected routinely at prenatal visits, and though largely un-modifiable, may aid in detecting risk and providing evidence for clinical decisions on who warrants prenatal depression screening to temporally monitor symptoms and the need for intervention. Future studies may want to examine how different prenatal cohort demographics in clinical settings serve as predictors of postpartum depression (PPD). Such investigations hold potential to leverage the use of existing data with large sample sizes to inform how routinely collected clinical data can be aggregated and translated into mechanisms for perinatal depression risk screening and provide evidence to inform clinical decisions in who to screen during pregnancy.

The support of a partner is commonly suggested to be a protective factor for perinatal depression, yet partner status was only reported in eight (32%) [39, 42, 50, 51, 55, 56, 58, 59] of the 25 studies. Further, three studies [42, 56, 58] had at least half of the sample comprised of single individuals, and four studies [50, 51, 56, 59] controlled for partner status in the analyses. Interestingly, only two studies [44, 55] reported the mode of delivery (8%), and four studies (16%) [43, 44, 53, 62] reported breastfeeding status. In the US, the overall cesarean delivery rate increased by 60% from 1996–2009 (20.7% to 32.9%), and then experienced a slight decline in 2019 (31.7%) before increasing again in 2020 (31.8%) and 2021 (32.1%) [69]. Though the COVID-19 pandemic may explain the most recent increase in cesarean deliveries, growing evidence indicates there are psychological consequences associated with cesarean deliveries, especially in the context of emergency cesarean deliveries [69] and for Black/African American delivering persons [70]. Regarding breastfeeding status, the direction and association of breastfeeding and perinatal depression has been controversial as some studies indicate breastfeeding as a protective factor [71]. Conversely, it has been indicated that perinatal depression may result in early cessation or that difficulties with breastfeeding may contribute to perinatal depression symptoms [72, 73]. Thus, mode of delivery and breastfeeding status may be important variables to consider in future investigations given the potential for psychological implications.

## Methodological factors

There were 9 prospective cohort studies [45, 47, 52, 54, 56–58, 60, 63], eight cross-sectional studies [40, 42, 44, 46, 49–51, 53], and six case-control studies [39, 41, 43, 55, 59, 62]. There was also one pilot study [48] and one randomized control trial [61]. The most common types of analytic methods applied were those looking at group differences (92%), correlations (52%), and regression (36%) while more complex forms of analyses, such as, mixed effects modeling (8%) and path analysis (2%) were the least common.

A total of four [44, 56–58] of the 25 studies discuss conducting an *a prior* power analysis to calculate the needed sample size with half of those being US based studies [48, 56]. However, of the two US based studies reporting a power analysis, one [48] does not report the calculated sample size nor if the study was sufficiently powered. Though power analysis was only reported in 16% of the 25 total studies, over half (60%) [43, 45–50, 52, 54, 55, 57, 60, 62, 63] note a small sample size as a study limitation. The percentage of US versus non-US based studies reporting sample size as a study limitation was equal at 60% each. We suspected secondary use of data from government or publicly available datasets with large sample sizes would explain the difference in sample sizes between US versus non-US based studies. Nearly half of the studies were secondary analysis (48%) [44, 46, 47, 49–53, 55, 56, 59, 63], but no studies explicitly reported the use of government or publicly available datasets.

All studies conducting biospecimen collection [39–46, 48, 49, 53–58, 60, 62, 63, 74] provided methods for processing and analyzing of the samples, though the level of detail provided was variable. All biospecimen samples were blood except for three studies that also collected either saliva [54], fecal [48], or urine [63] samples in addition to blood samples. A total of six [39, 41, 42, 56, 58, 62] studies reported the time of biospecimen collection, and one study reported requiring fasting (12 hours) when collecting blood samples [41].

Though not unexpected, the Edinburgh Postnatal Depression Scale (EPDS) was the most used instrument to measure depression (60%) [39, 41, 43–45, 48, 52–55, 57, 58, 60, 62, 63] followed by the Center for Epidemiological Studies Depression (CES-D) (16%) [42, 46, 50, 61]. Only 20% of the studies [47–49, 56, 59] utilized semi-structured interviews to measure depression for purposes other than group allocation (i.e., depressive, control) and/or study eligibility. Of the 15 studies using EPDS to measure depression, 40% did not report a specified cut-off score [45, 48, 53, 55, 57, 63]. The nine studies [39, 41, 43, 44, 52, 54, 58, 60, 62] reporting EPDS cut-off scores varied between 9–13. The most common cut-off score was 10 (33.3%) [44, 54, 62], which is lower than the current clinically recommended cut-off score of ≥13 [75]. Of the seven studies [41, 43, 44, 52, 54, 60, 62] using an EPDS cut-off score other than ≥13, only three studies (28.6%) [41, 43, 52] provided scientifically supported rationales for using an alternative cut-off score. The US Preventive Services Task Force indicated 10 and 13 as the most common cut-off scores used [10]. A recent individual participant data meta-analysis suggested that using a cut-off score of ≥ 11 may be preferable due to combined sensitivity and specificity being maximized [76, 77]. However, the current recommendation of ≥13 has remained unchanged since it was developed by Cox and colleagues (1987) nearly four decades ago in a postpartum sample in the United Kingdom.

As determined by quality appraisal assessments, concerns for risk of bias were noted or were unclear related to the following types of bias: 13 (52%) selection bias, 3 (12%) recall bias, and 24 (96%) measurement bias (**Fig 2**). A narrative description of risk of bias considerations for each included article is detailed in **S1 Table**. Moderate level of risk was noted in six studies for selection bias [41, 46, 47, 49, 51, 53], one for recall bias [46], and zero for measurement bias, whereas high level of risk was noted in two studies for selection bias [50, 52], two for recall bias [59, 63], and two for measurement bias [55, 59]. Further, a majority of the studies (88%) were indicated as having an unclear risk of bias for measurement bias largely due to studies not providing sufficient information or references to support the use of the measurement with respect to their sample characteristics and/or cut-off scores. For instance, Sha and colleagues (2022) conducted a study in a non-Swedish sample (US based sample) but referenced a study validating the Swedish version of the EPDS in pregnancy. Another example is that of Miller and colleagues (2018) who used CESD to measure perinatal depression, and their supporting reference was a study assessing the efficacy of the instrument for use as screener for depression in community residing older adults (50–96 years of age).

Further, it is important to note the items comprising the EPDS were adopted from existing scales mainly developed in the United Kingdom (UK) in non-perinatal populations of variable age (16–65) [75, 78–80]. The sample characteristics described by Cox and colleagues (1987) are incongruent with all 15 studies that reported using the EPDS. Though there are notable differences in sample characteristics (i.e., country, mode of delivery, social class, relationship status, language), a total of 10 (66.7%) studies [39, 43, 47, 48, 52–57] cite Cox and colleagues (1987) with 60% of these studies [39, 43, 48, 55–57] using this reference to substantiate the validity and reliability of the instrument and/or cut-off score for use in their study. The EPDS is currently considered "gold standard" for measuring perinatal depression, but increased inclusion of supporting references and/or scientifically supported rationales may be

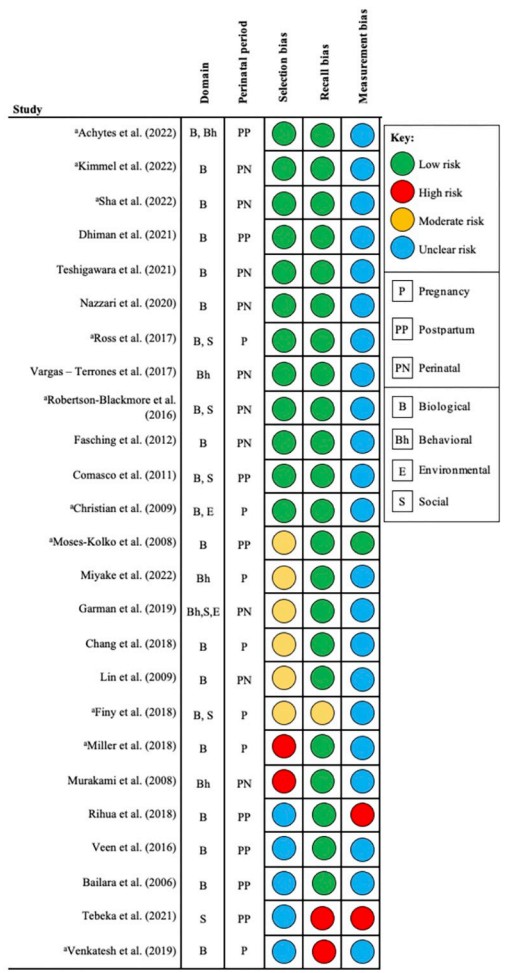

**Fig 2. Summary of level of risk of bias per study.** [a]US based study.

particularly useful to aid in decreasing variability in cut-off scores by collectively establishing best practices for determining cut-off scores respective to sample characteristics.

**Methodological considerations.** As demonstrated in this review, insufficient evidence is being provided for instrument selection in measuring perinatal depression. Without robust measures for primary outcome or group allocation variables, the risk of compromising the integrity of subsequent findings and the wider body of evidence is high. While widely used instruments like the EPDS have remained unchanged for decades, social and political norms for child-bearing persons and marginalized groups have evolved since the 1980's. Thus, the psychometric properties of these instruments warrant continuous critical examination, especially when being used in diverse samples. It is equally important supporting references for instrument selection, respective to sample characteristics, are reported to generate evidence for instrument validity and reliability across diverse samples, to establish best practices, and indicate when modifications and/or the development of new measures may be warranted.

Given the variability in cut-off scores and evidence suggesting perinatal depression may phenotypically differ between pregnancy and postpartum as well as from that of non-

reproductive depression [81, 82], future investigations may want to consider the utility of existing perinatal depression measures for present day use. Such investigations could aid in determining if modifications are needed to improve the scientific and clinical utility of perinatal depression measures. While we acknowledge the limitations of incorporating clinical interviews as a form of data collection (e.g., time constraints, burdensome to participants/staff, training, internal validity concerns), future investigations may be strengthened by conducting semi-structured interviews, in addition to self-report measures, when measuring perinatal depression. Incorporating two forms of measurement that yield two types of data (i.e., qualitative, quantitative) may strengthen any subsequent findings and progress our understanding of depression symptoms exclusive to perinatal populations. Progress of this nature could lead to advancements in life-stage informed measures that increase precision in detection and timely intervention.

The bioavailability of essential amino acids (e.g., tryptophan, competitor amino acids), the precursors to a number of neurotransmitters commonly associated with psychiatric conditions, depends on dietary intake [24, 83]. Thus, biospecimen collection, as it relates to timing of food consumption, is likely important to consider in investigations including essential amino acids since levels may significantly vary depending on when sample collection takes place. Yet no studies reported biospecimen collection time in relation to timing of food consumption suggesting this is not common practice. Free (non-albumin bound) tryptophan (TRP) is what can be transported across the blood-brain-barrier (BBB) to make it available in the brain for serotonin synthesis [24]. Conversely, it has been suggested that TRP has a higher affinity for the BBB than for albumin, and albumin bound TRP close to the BBB may separate from albumin to then transport across the BBB [24, 84, 85]. Meaning, measurement of both free and total TRP is likely important in the study of psychiatric conditions, but only one study [40] specified if free and/or total TRP was measured. Future investigations including essential amino acids may want to 1) consider biospecimen collection times in relation to timing of food consumption to advance our understanding of tryptophan metabolism in the perinatal period and 2) to clarify if free and/or total TRP is being measured as such considerations are essential for making meaningful interpretations of the findings.

Lastly, each type of biospecimen and method for processing and analyzing of samples introduces bias innate to the specified type and method [86]. Therefore, decisions on what type of biospecimen(s) to collect and methods of analysis warrant thoughtful consideration. Findings from this review suggest a need for increased transparency in reporting of methods and rationales to support methodological decisions. Transparency is vital not only for the purposes of reproducibility but also to collectively establish best practices for methods of biological sample selection, collection, processing, and analysis. Overall, these findings suggest that methods of investigation in maternal mental health science have room for improvement and can be strengthened with increased attention and reporting of sufficiently supported methodological decisions and processes, such as those discussed in this review. By strengthening the methods of investigation in maternal mental health science, we can progress standards for best practices as well as mitigate the risk of generating conflicting findings that are a result of unsound methods rather than true conflicting findings.

**Contextual considerations.**   Since federal funding (e.g., National Institutes of Health [NIH]) is one of the largest sources of research funding [87], we evaluated the scope of federal funding allocated to maternal depression research to further address our final aim. To better understand potential system level factors that may partly explain the methodological findings discussed in this review, we examined the number of publications on maternal depression resulting from federally funded projects and the number of projects that have been funded on

maternal depression to date. An in-depth analysis respective to research funding is beyond the scope of the present review. Thus, future research is warranted.

We used NIH RePORTER (reporter.nih.gov) to approximate the number of publications funded by NIH projects on "maternal depression" and the number of federally funded projects where "maternal depression" was a primary project focus. Steps for this process can be seen in **Fig 3**. The publication search yielded 36,425 publications supported by 906 core projects (1985–2023). Once duplicate publications and publications not specific to maternal depression were removed (i.e., infant outcomes, other non-perinatal population outcomes), only 267 (0.7%) publications under 192 (21.2%) core projects (1991–2022) and 131 (0.36%) non-US based publications under 99 (10.9%) core projects (2002–2022) remained from the initial search total. The project search for "maternal depression" initially resulted in 3,488 projects (1985–2023). Similar to publications, once duplicate projects and projects with primary outcomes not specific to perinatal persons were removed, only 158 projects (4.5%) spanning over 38 years remained. Of the 158 projects, 92 (58.2%) were intervention studies. Further, a number of southern states have some of the highest maternal mortality rates and/or poor maternal mental health outcomes yet were among the lowest funded states for investigations on maternal depression (e.g., Louisiana (0), Arkansas (0), Mississippi (0), New Mexico (0), Kentucky (1), Texas (4)) [1, 8, 64, 88, 89].

These findings establish the first federally funded project on perinatal depression began 38 years ago indicating perinatal mental health is a relatively new area of investigation, yet temporal trends in funding appear to be partial to intervention-based studies. Further, the amount of funding allocated to maternal depression research may be overinflated once accounting for funding that does not have maternal outcomes as a primary focus. Though further investigation is warranted, barriers to funding for maternal mental health focused projects may partly explain why maternal mental health scientists are largely relying on secondary use of data to generate new knowledge.

Secondary use of data inevitably limits study design and methodological decisions. However, timely advancements in maternal mental health science and care are needed as every human develops within a maternal environment (womb) for up to 10 months and maternal mortality and morbidity rates continue to rise [1, 64, 90–92]. Therefore, it is imperative that maternal mental health gains recognition as a public health issue and sources of funding begin to prioritize maternal mental health science and care, especially in those states with higher disease burden and mortality rates [90].

## Biological determinants

A total of 20 studies [39–46, 48–50, 53–58, 60, 62, 63] investigated biological determinants of perinatal depression. Inflammatory markers were investigated in 10 studies [39, 41, 42, 46, 50, 54, 56–58, 63], tryptophan and/or tryptophan metabolites in seven studies [39, 40, 48, 55, 58, 60, 62], genetic polymorphisms in three studies [43, 45, 49], micronutrient alterations in two studies [41, 44], and neurological factors in one [53], respectively. A summary of findings with statistical values for each of the 20 studies can be found in **Table 2**.

**Inflammatory markers and oxidative stress.** TNF-α (pro-inflammatory cytokine) was positively correlated with prenatal depression and those with prenatal depression had higher TNF-α levels compared to those without [41, 50]. Miller and colleagues (2018) found that even when controlling for sociodemographic factors, those with prenatal depression unresponsive to antidepressant treatment and those with untreated prenatal depression had higher TNF- α levels compared to those with prenatal depression that responded to antidepressant treatment. These findings suggest that TNF-α may be a useful biomarker for determining a subtype of perinatal depression that is treatment resistant to antidepressants. However, it is important to

## NIH RePORTER (Publications)

### Step 1
*Search "maternal depression"*

### Step 2
*Export data*
Publications (N = 36,425)
Projects (N = 906)

### Step 3
*Remove duplicates, publications not specific to perinatal persons*
Publications (N = 267)
Projects (N = 192)
1991-2022

## NIH RePORTER (Projects)

### Step 1
*Search "maternal depression"*

### Step 2
*Export data*
Projects (N = 3,488)

### Step 3
*Remove duplicates, publications not specific to perinatal persons*
Projects (N = 158)

### Step 4
*Organize data by state column*
At the state level, projects were then quantified to assess funding in states with some of the highest maternal mortality and/or maternal mental health rates
Lousiana (n = 0); Arkansas (n = 0); Mississippi (n = 0); New Mexico (n = 0); Kentucky (n = 1); Texas (n = 4)

### Step 5
*Quantify intervention studies*
*(N = 92)*
*Keywords searched (intervention, treatment, therapy, therapeutic)

**Fig 3. Steps for NIH RePORTER data acquisition.**

note Miller and colleagues (2018) do not specify specific antidepressants used for treatment nor the duration of treatment. Additionally, intimate partner violence is commonly indicated as a risk factor for perinatal depression. Robertson-Blackmore and colleagues (2016) found a history of intimate partner violence to be positively associated with TNF-α. Also suggesting

**Table 2. Summary of findings—biological determinants.**

| Study | Purpose/Aims | Design / Timepoints | Determinants (from other domain) | Summary of findings (values) |
|---|---|---|---|---|
| **PREGNANCY (Biological)** | | | | |
| ®Venkatesh[2] (2019) (N = 462) US | 1) Determine whether antenatal depression was associated with two biomarkers of oxidative stress, 8-OHdG and 8-Isoprostane, and five biomarkers of inflammation. 2) assess whether the association between antenatal depression and SPTB was mediated by those biomarkers found to be significant in the primary aim. | Prospective 10-, 18-, and 26-weeks gestations | Inflammatory markers; oxidative stress | **Spontaneous preterm birth (SPTB)** was *2 times more frequent* among those with depression compared to those without (*12.4 vs. 6.3%, OR: 2.1 [95% CI: 1.10–4.04], p = 0.02*) <br><br> Those with depression had ↑ levels of specific gravity corrected **8-isoprostane** compared to those without depression (*geometric mean: 299.96 pg/mL vs. 237.01 pg/mL, p = 0.001*). <br><br> Those with depression who had prenatal antidepressant exposure had ↓ levels of **8-isoprostane** compared to those who had depression without antidepressant exposure (*geometric mean: 362.40 pg/mL, p = 0.03*); however, both groups (antidepressant exposure vs. not) had ↑ **8-isoprostane** levels compared to those without prenatal depression (*237.01 pg/mL, ANOVA p = 0.02*). <br><br> Prenatal depression was associated with **SPTB** (*AOR: 2.09, 95% CI: 1.09–4.03, p = 0.02*). The association between **8-isoprostane** and prenatal depression with **STPB** were ↓ when analyzed in the same *regression model*, which is suggested by the authors to indicate *partial mediation* of **8-isoprostane** on the relationship between prenatal depression and **SPTB** (*AOR for 8-isoprostane: 3.72, 95% CI: 2.14–6.46, p < 0.001; AOR prenatal depression: 1.68, 95% CI: 0.85–3.34, p = 0.13*). <br><br> After *bootstrapping* over 1,000 iterations, it was found that 27% of the effect of prenatal depression on **SPTB** was explained by **8-isoprostane.** <br><br> No significant findings were noted for **8-OHdG** or inflammatory markers. |
| ®Finy[2] (2018) (N = 214) US | Examine the association between childhood abuse, low socioeconomic status (SES) and inflammatory markers during pregnancy | Cross-sectional ≤ 31 weeks gestation or ~30 days after flu vaccination | Inflammatory markers (social support, significant life events) | **Childhood abuse history** was *positively associated* with CRP and IL-6. **Current SES** and CRP and **IL-6** were *negatively associated (p's < 0.01)*. <br><br> Depressive symptoms were *positively correlated* with **IL-6** (*r = 0.23, p < 0.01*). |
| ®Miller[2] (2018) (N = 170) US | To evaluate the association between psychotropic medication and maternal serum inflammatory biomarkers in women with antenatal depressive symptoms (ADS) in the mid-trimester. | Cross-sectional 12–21 weeks gestation | Inflammatory markers | Those with underlined untreated depression were more likely to be from a racial/ethnic minority group, to have a ↓ household income, to be publicly insured, have a ↓ educational level, and ↓ likely to be married. Further, they were ↑ likely to be employed than those with depression non-responsive to treatment but were ↓ likely to be employed than those with depression responsive to treatment. <br><br> There were no differences noted in serum levels of IFNγ, IL13, IL6, IL8, or CRP, but TNF-α differed across the groups. *Post-hoc* analyses indicated those non-responsive to treatment (p = 0.02) and untreated depression (p = 0.01) had ↓ TNF-α compared to those responsive to treatment. <br><br> No differences noted between untreated depression and those non-responsive to treatment (p = 0.76). <br><br> When controlling for race/ethnicity, income, and marital status, a *linear regression* demonstrated both those with depression who were non-responsive to treatment and those who had untreated depression had ↑ TNF-α compared to those responsive to treatment (β = 0.27, 95% CI: 0.02–0.52 and β = 0.23, 95% CI 0.02–0.44). |
| Chang (2018) (N = 33) Taiwan | Investigate if subjects with depression in pregnancy had higher levels of pro–inflammatory markers and lower levels of anti-inflammatory markers. | Case control 16–28 weeks gestation | Inflammatory markers; micronutrient alterations | Compared to controls, those with prenatal depression had ↓ levels of **omega-3 polyunsaturated fatty acid (3-PUFAs)** (*p = 0.026*), **EPA** (*p = 0.019*), and **DHA** (*p = 0.02*). They also had ↑ **n-6/n-3 ratios.** <br><br> **TNF-α** was the only **inflammatory marker** found to be significantly ↑ for those with prenatal depression versus those without (*p = 0.016*). <br><br> No *correlation* between depression severity PUFAs and **inflammatory markers** were found. Depression duration was *negatively correlated* with **total n-3 PUFAs, EPA** and **DHA** (*r = –0.415, –0.395, –0.392, p = < 0.05*). Current depression was *positively correlated* with **n-6/n-3 ratio** and **TNF-α** (*r = 0.458, 0.443, p < 0.01*). |

(Continued)

**Table 2.** (Continued)

| Study | Aim | Design / Timing | Determinant | Findings |
|---|---|---|---|---|
| ® Ross (2017) (N = 90) US | Examine the association between pregnant women's close relationships and cytokine profiles in the third trimester. | Prospective 22–26 and 32–36 weeks gestation | Inflammatory markers (social support) | *Correlations between cytokines varied within each trimester and ranged from $r = 0.660$ –$r = –0.469$ with a mean $r = 0.322$ indicating a good proportion of variance in each cytokine is unique.* **Romantic partner (RP) relationships** with **positive features (i.e., support/closeness)** were associated with ↓ levels of **inflammatory cytokines; RP relationships low** in both **positive and negative features (indifferent)** were *associated with cytokine profiles* indicating ↑ **inflammation.** **Positive RP** relationship was negatively associated with **IL6:IL10 ratio.** Further, when **positive RP** features were ↑ and there were ↓ **RP negative** features, the estimated **IL6:IL10 ratios** were lowest indicating a potential buffering or protective effect of **positive RP relationships.** **Positive** and **negative RP relationships** were *associated* with **IL10** levels ($b(SE) = 0.031\ (0.009), p = 0.001$; $b(SE) = 0.017\ (0.007), p = 0.017$). **Positive** and **negative RP relationships** were *associated* with **IFNγ** levels ($b(SE) = 0.131\ (0.041), p = 0.002$; $b(SE) = 0.095\ (0.032), p = 0.004$) Neither **positive** and **negative RP relationships** were *associated* with **IL13, IL8, IL6,** and **TNF-α** levels. ↑ **positive RP relationship** was *associated* with ↓ depressed mood ($r = –0.35, p = 0.001$) and **perceived stress** ($r = –0.41, p < 0.001$) whereas ↑ **negative RP relationship** was *associated* with ↑ depressed mood ($r = 0.51, p < 0.001$), **perceived stress** ($r = 0.53, p < 0.001$), and **pregnancy distress** ($r = 0.29, = 0.005$). |
| ® Christian (2009) (N = 60) US | Examine associations among perceived stress, current depressive symptoms, and serum inflammatory markers among pregnant women from primarily lower socioeconomic backgrounds. | Cross-sectional First and second trimester | Inflammatory markers (social support, perceived stress) | When controlling for **pre-pregnancy BMI**, ↑ depression scores were *associated* with ↑ levels of **IL-6** ($\beta = .23, t(2, 55) = 1.98, p = 0.05$). ↑ depression scores were marginally *associated* with ↑ **TNF-α** levels ($\beta = 0.24, t(2, 58) = 1.06, p = 0.06$). |

**POSTPARTUM (Biological)**

| Study | Aim | Design / Timing | Determinant | Findings |
|---|---|---|---|---|
| Dhiman[2] (2021) (N = 660) India | Explore the association between vitamin B12 and probable PPD in South Indian population. | Cross-sectional 6 weeks postpartum | Micronutrient alterations (dietary intake) | Median **total B12** levels and **cB12** were ↓ in cases compared to controls ($p < 0.001$). **Methyl malonic acid (MMA)**–marker of functional deficiency of vitamin B12 –was ↑ cases compared to controls ($p = 0.002$). After *adjusting* for **SES, martial dissatisfaction, unplanned pregnancy, and type of delivery,** the *regression model* indicated the likelihood of postpartum depression to ↓ by 0.39 for ever unit ↑ in total **vitamin B12** ($OR = 0.394$; 95% CI: 0.189–0.822, $p = 0.009$) and by a factor of 0.29 ($OR = 0.293$; 95% CI: 0.182–0.470, $p < 0.001$) for **cB12. MMA** ($OR = 2.04$, 95% CI: 1.53–2.11, $p < 0.001$) and **5-methyl tetrahydrofolate (THF)** ($OR = 3.18$; 95% CI: 1.42–6.08, $p = 0.001$) were found to be predictors of PPD. After *adjusting* for **SES, martial dissatisfaction, unplanned pregnancy, and type of delivery,** a significant *negative association* among **serotonin** and depression remained ($\beta = –0.16, p = 0.005$), as did a *positive association* among **MMA** ($\beta = 0.161, p = 0.001$), **homocysteine (hcy)** ($\beta = 0.155, p = 0.005$), and **THF** ($\beta = 0.118, p = 0.010$) and depression. The *path analysis* model with total **vitamin B12** as the predictor, depression score as the outcome variable, and **MMA** as the *mediator* was significant ($p < 0.001$). |

*(Continued)*

**Table 2.** (Continued)

| Study | Aim | Focus | Design/Timepoint | Results |
|---|---|---|---|---|
| Achtyes (2020) (N = 130) US | Investigate whether a pro-inflammatory status in plasma, together with changes in the kynurenine pathway activity, is associated with the development of severe depression and suicidal behavior in the post-partum. | Inflammatory markers; tryptophan pathway (suicide) | Case-control 6–12 weeks | ↑ **IL-6, IL-8** ↑ PPD (*OR IL-6 = 3.0, 95% CI = 1.37–6.6; OR IL-8 = 3.32, 95% CI = 1.32–8.34, per pg/ml increase*) <br> ↓ **IL-2** ↑ PPD (*OR = 2.34, 95%CI = 1.35–4.05, p = 0.002, per pg/ml decrease*) <br> ↓ **serotonin** ↑ *odds* of PPD (*OR = 1.43 per nM decrease in serotonin, 95% CI: 1.07–1.92, p = 0.016*) <br> ↑ **Kynurenine/serotonin** ratio ↑ PPD (*OR = 1.35 per unit increase, 95% CI: 1.03–1.79, p = 0.038*) <br> *Sensitivity analysis* using depression scores: models for **IL-8, IL2, serotonin, serotonin/kynurenine,** and **quinolinic acid** were significant; *(linear regression, Beta 3.9, Standardized Beta 0.22, p = 0.006), (linear regression, Beta −2.3, Standardized Beta −0.23, p = 0.005), (linear regression, Beta −1.3, Standardized Beta −0.24, p = 0.003), linear regression, Beta −1.1, Standardized Beta 0.22, p = 0.009), linear regression, Beta −4.3, Standardized Beta −0.18, p = 0.022)* |
| Rihua[2] (2018) (N = 84) China | To determine associations between PPD and plasma neurotransmitters. | Tryptophan pathway | Case control *2 weeks postpartum* | There were significant differences in **education** and **mode of delivery** among those with PPD and those without. <br> Plasma levels of **serotonin (5-hydroxytryptamine or 5-HT)** and **neuropeptide Y (NPY)** were ↓ in those with PPD compared to controls (*p < 0.05 or p < 0.01*) whereas **norepinephrine (NE)** and **substance P (SP)** were ↑ in PPD cases versus controls (*p < 0.05*). No differences were found for **dopamine (DA).** <br> A *negative correlation* among depression scores and **serotonin** and **NPY** (*p < 0.05 or p < 0.01*) were present as well as a *positive correlation* among depression scores with **NE** and **SP** (*p < 0.01 or p < 0.01*). <br> ↓ **serotonin** was *associated* with **current** and **history of suicidal behavior** and ↑ *odds* of **completed suicide** attempt during pregnancy. (*OR: 0.51[0.32, 0.8]1, p = 0.005), (OR: 0.50 [0.29, 0.87], p = 0.013), (OR: 0.51, [0.31, 0.84], p = 0.007*) |
| Veen (2016) (N = 42) Netherlands | To investigate if alterations in tryptophan degradation in the postpartum period are associated with the occurrence of postpartum depression and postpartum psychosis. | Tryptophan pathway | Case control Postpartum timepoints not specified | Those considered to be "healthy" postpartum participants were ↑ likely to be **breastfeeding** at the time of blood collection (*p < 0.001*). <br> ***Physiological postpartum period:*** <br> Healthy postpartum (PP) participants had ↓ serum levels of **kynurenic acid (KA)** compared to healthy non-PP controls (*p < 0.001*). <br> All PP participants had ↑ levels of **3-OH-kynurenine (3HK)** (*p = 0.011*); the **KA/kynurenine (KYN) ratio** was ↓ in healthy PP participants (*p < 0.001*) suggesting a strong inhibition of the **kynurenine aminotransferases (KAT) enzymes** during the first 2 months PP. <br> The **3HK/KYN ratio** was ↑ in healthy PP participants with a median time of 22 days PP (*p = 0.021*), but not in healthy PP participants with a median time of blood collection 40 days PP. The authors suggest that this indicates ↑ activity of the **kynurenine-3-monooxygenase (KMO) enzymes** in the first month of the physiological PP period and then the gradual returning to "normal" levels. <br> The **serotonergic pathway (5HIAA)/KYN ratio** was ↓ in healthy PP participants suggesting that the breakdown of **tryptophan (TRP)** is *biased towards* the KYN pathway and *away from the* **serotonergic** pathway in the physiological PP period (*p = 0.009*). <br> "Healthy" PP participants had ↓ serum levels of **TRP** (*p < 0.001*), and ↑ levels of **KYN** (*p = 0.002*) compared to healthy non-PP participants, and consequently the **TRP breakdown index** was also ↑ (*p < 0.001*). <br> **KYN** was ↓ in cases compared to controls (*p = 0.001*), and accordingly cases had a ↓ **tryptophan breakdown index** compared to controls (*p = 0.035*). |

*(Continued)*

**Table 2.** (Continued)

| Study | Aim | Study design | Factor | Findings |
|---|---|---|---|---|
| Comasco (2011) (N = 272) Sweden | Examine whether genetic variations in the monoaminergic neurotransmitter system, together with environmental stressors, contribute to the development of PPD symptoms | Case control 6 weeks and 6 months postpartum | Genetic polymorphisms (significant life events, social support, stress, unhappiness with pregnancy) | *Associations between **genetic polymorphisms** and PPD symptoms were significant only at the 6-week time point, not at 6 months. However, Comasco and colleagues (2011) did not provide statistical output for these associations."* <br><br> **COMT-Val**[158]**Met** with ↑ risk for **Met carriers** was *associated with PPD.* <br><br> Gene-by-gene interactions were present for **COMT-MAOA** in relation to PPD symptoms. **Low MAOA activity carriers** with the **Met variant of COMT** was related to PPD symptoms; **high MAOA activity variant** was *associated* with PPD symptoms only when **combined** with the **Met allele of COMT; short 5HTT allele** was *associated* with PPD symptoms only when **combined** with the **Met allele of COMT.** <br><br> **COMTVal**[158]**Met** was *associated* with PPD symptoms in the presence of **previous psychiatric contact** and **maternity stressors**, while **MAOA-uVNTR** was *associated* with PPD symptoms only in the presence of maternity stressors. <br><br> The *logistic regression analysis demonstrated an association among PPD symptoms and **COMTVal**[158]**Met, previous psychiatric contact,** and **maternity stressors**. The model explained 30% variance.* After stratifying for previous psychiatric contact, the gene-environment interaction model indicated those with **previous psychiatric contact** had a *main effect* of **COMT-Val**[158]**Met** and **5HTT-LPR** with an *explained variance of 40%.* |
| Moses-Kolko[2] (2008) (N = 16) US | To measure brain serotonin-1A (5HT1A) receptor binding potential (BP) in healthy and depressed postpartum women. | Cross sectional ≤ 16 weeks postpartum | Neurological factors | There was an effect of **breastfeeding status** on **hypothalamic-pituitary-ovarian axis** hormone concentrations **estradiol, progesterone, LH, FSH,** and **prolactin** [*Wilks' lambda = 0.2056; $F_{(5, 10)} = 7.73$, $p = 0.003$*]. <br><br> A *post-hoc analysis showed **breastfeeding** was associated* with ↓ **estradiol** [$F_{(1, 14)} = 8.31$, $p = 0.01$], **progesterone** [$F_{(1, 14)} = 4.33$, $p = 0.06$], and **FSH concentrations** [$F_{(1, 14)} = 5.18$, $= 0.04$] and ↑ **prolactin concentrations** [$F_{(1, 14)} = 26.25$, $p = 0.0002$]. <br><br> **Serotonin receptor (5HT1A) binding** in the three *a prior* regions of interest (**mesiotemporal cortex, left lateral orbitofrontal cortex, and subgenual anterior cingulate cortex**) demonstrated a *main effect of* depression [$F_{(3, 12)} = 13.67$, *Wilks' lambda = 0.23*, $p = 0.0004$]. <br><br> *Post hoc analysis detected significant depression effects on ↓* in the **mesiotemporal cortex** [*21.6% mean decrease;* $F_{(1, 140} = 22.5$, $p = 0.0003$], **subgenual cingulate cortex** [*27.65 mean decrease;* $F_{(1, 14)} = 23.4$, $p = 0.0002$], and **left lateral orbitofrontal cortex** [*17.9% mean decrease;* $F_{(1, 14)} = 7.13$, $p = 0.018$] regions. There were also associations with reductions in the secondary ROI [$F_{(5, 10)} = 3.24$, *Wilks' lambda = 0.38*, $p = 0.054$], and the most significant ↓ were in the **right lateral orbitofrontal cortex** [*23.4% mean decrease;* $F_{(1, 14)} = 8.72$, $p = 0.011$] and **pregenal anterior cingulate cortex** [*23.4% mean decrease;* $F_{(1,14)} = 17.2$, $p = 0.001$]. |
| Bailara (2006) (N = 50) France | Assess the correlation of intensity of baby blues, with the intensity of metabolic changes and brain tryptophan availability | Cross-sectional *"just before delivery"* (baseline) and three days after delivery. | Tryptophan pathway | Total plasma **TRP** exhibited a *mild (+19%)* ↑. <br><br> An abrupt ↑ in **competitor amino acid** concentrations (+77% **isoleucine**, +55% **leucine**, +52% **tyrosine**) led to a ↓ in **brain tryptophan availability (BTAI).** <br><br> The **BTAI** ↓ between the *prenatal and postpartum period (-15%, $p < 0.01$) and was associated with PP blues symptoms.* <br><br> The change in **BTAI** was *negatively correlated* with the intensity of postpartum blues (*r = -0.283, $p < 0.05$*). |
| | | | **PERINATAL (Biological)** | |

*(Continued)*

**Table 2.** (Continued)

| Study | Aim | Design/Timing | Determinants | Findings |
|---|---|---|---|---|
| ®Kimmel (2022) (N = 30) US | Analyze trajectories of serotonin and tryptophan-related metabolites, bile acid metabolites, and microbial composition related to psychiatric history and current symptoms across the perinatal period. | Pilot First or second trimester; 32–37 weeks gestation; 5–10 weeks postpartum | Tryptophan pathway | Mean **serotonin** level ↑ from pregnancy to postpartum ($p = 0.0002$ for $3^{rd}$ trimester (V2) to 5–10 weeks postpartum (V3); $p = 0.002$ for $1^{st}$ or $2^{nd}$ trimester (V1) to V3). **NEOP level** trajectories followed a different pattern than **serotonin** by ↑ from V1 to V2 ($p < 0.0001$) and then ↓ postpartum ($p = 0.005$). Mean **KYN** ↑ from V1 to V2 ($p = 0.003$) and ↑ again from V2 to V3 ($p = 0.004$). The **KYN/TRP** ratio was ↑ at V2 and V3 compared to V1 ($p < 0.0001$; $p < 0.0001$). **KA** was ↑ at V3 compared to both V2 ($p = 0.003$) and V1 ($p = 0.0004$).<br><br>**Primary bile acids:**<br>**Chenodexycholic acid (CDCA)** ↑ from V2 to V3 ($p < 0.00011$) with an overall ↑ from earlier V1 to V3 ($p = 0.0003$); **Glycochenodeoxycholic acid (GCDCA)** ↑ from V2 to V3 ($p < 0.0001$) and remained ↑ at V3 compared to V1 ($p < 0.0001$); **Taurochenodeoxycholate (TCDCA)** ↓ from V2 to V3 ($p = 0.001$); **Glycocholic acid (GCA)** ↑ from V1 to V2 ($p = 0.003$) and ↑ from V1 to V3 ($p = 0.005$); **Taurocholic acid (TCCA)** ↑ from V1 to V2 ($p < 0.0001$), and ↓ from V2 to V3 ($p < 0.0001$).<br><br>**Secondary bile acids:**<br>**Glycoursodeoxcholic acid (GUDCA)** ↓ from V2 to V3 ($p < 0.0001$) whereas **GUDCA** and **Ursodeoxycholic acid (UDCA)** ↑ from V2 to V3 ($p < 0.0001$; $p = 0.0003$) and **GUDCA** remained ↑ from V1 to V3 ($p < 0.0001$); **Glycolithocholic acid (GLCA)** ↑ from V2 to V3 ($p < 0.0001$) and levels of **Glychoyocholic acid (GHCA)** and **GLCA** were ↑ compared to the initial value in pregnancy ($p = 0.0001$; $p < 0.0001$; $p = 0.0005$)<br>**Tauro alpha-murcholic acid (TaMCA), Taurohyocholic acid (THCA),** and **tarodeoxycholate hydrate (TDCA)** ↓ from V2 to V3 ($p < 0.0001$; $p = 0.0003$; $p < 0.0001$) and V3 were ↓ than earlier in pregnancy ($p < 0.0001$; $p = 0.0003$; $p = 0.002$).<br>**TUDCA, TDCA,** and **TCA** were *associated* with change in **NEOP** from V1 to V2 ($q = 0.011$; $q = 0.021$; $q = 0.021$). **TUDCA** was also *associated* with change in **TRP** ($q = 0.004$), **KYN** ($q = 0.001$), and **KA/KYN** ratio ($q = 0.002$). These findings became stronger when excluding those in the first trimester.<br><br>**Metabolites and microbiome:**<br>**Alpha diversity** did not significantly change across the perinatal period. ↑ **bile acid GUDCA** and **UDCA** levels were *associated* with ↓ **alpha-diversity** across all 4 indices *(evenness, Faith's phylogenetic diversity, count of observed OTUs, Shannon entropy).*<br>↑ **CDCA** was *associated* with ↓ **alpha diversity** for the *evenness index* and *Shannon index* only, and also only when first trimester participants were included.<br>Certain **bacterial genera** were *associated* with **UDCA** and **TUDCA**, primarily in the order **Clostridiales** and family **Cachnospiraceae**. **THcA** was also *associated* with **Riseburia**. **UDCA** was the only metabolite associated with psychiatric history ($q = 0.033$). |
| ®Sha (2022) (N = 114) US | To determine whether cytokines and kynurenine metabolites can predict the development of depression in pregnancy. | Prospective First–Third Trimester and unspecified postpartum timepoint | Inflammatory markers; tryptophan pathway | ↑ **IL-1β, IL-6,** and **QUIN** were *associated* with ↑ depression severity and/or ↑ *odds* of having depression *(Percent change in OR(CI): 32.3% (7.0, 63.6), 58.4% (22.1, 111.7), 91.6% (15.0, 232.0)*<br>**IL-6** performed best in *predicting* depressive symptoms; however, **KYN, QUIN, KYN/TRP ratio (rKT)** also produced good predictions *(AUC = 0.79 and 0.8 by Bayesian ordinal and logistic regression, respectively; ROC AUC >0.7). Precision recall analyses confirmed predictive value of model.*<br>The *leave-one-out cross validation* method indicated the predictability of the model would be optimal from mid- to late pregnancy ($2^{nd}$ to $3^{rd}$ trimester). The **full model** nominally outperformed **individual markers** for *predicting* risk of significant depressive symptoms. *Ordinal and logistic regression full models had ROC AUC = 0.83, PR AUC = 0.41.* |

*(Continued)*

**Table 2.** (Continued)

| Study | Aim | Design | Category | Results |
|---|---|---|---|---|
| Nazzari (2020) (N = 97) Italy | 1) Describe the cross-sectional and longitudinal association between tryptophan, kynurenine, and kynurenine/ tryptophan ratio and depression symptoms in late pregnancy through the first year postpartum 2) examine the role of inflammatory (IL-6) and stress (cortisol) markers in moderating any associations 3) determine if specific to depressive symptoms or can be replicated with anxiety given high concurrence of these disorders | Prospective Biological markers were collected at 34–36 gestational weeks; other measures assessed "during pregnancy", 2 days postpartum, and 3 and 12 months postpartum. | Inflammatory markers | ↑ prenatal **Kyn** levels were *associated* with ↓ depressive symptoms in late pregnancy (*estimate = - 0.002, SE = 0.001, p = 0.03*) after adjusting for maternal age. **Pre-pregnancy BMI** was mildly *associated* with **IL-6** levels (*r = 0.23, p = 0.03*) in preliminary analysis but adjusting models for **BMI** did not alter the direction or significance of findings. *Model 2:* There was a *three-way interaction* among prenatal **Trp levels, IL-6,** and slopes of **time** on depression scores (*ps < 0.05*). ↓ levels of prenatal **Trp** and ↑ **IL-6** were *associated* with ↑ depressive symptoms in late pregnancy (*p = 0.04*) and with the change in depressive symptoms from pregnancy to three postpartum time points (*ps = 0.04*). *Model 3:* A *three-way interaction* among the **KYN/TRP ratio, IL-6,** and the depression scores trajectory from pregnancy to 12 months postpartum. ↓ levels of prenatal **KYN/TRP ratio** and ↑ levels of **IL-6** were *associated* with ↑ depressive scores at delivery (*p = 0.05*) and 12 months postpartum (*p = 0.004*) and with a flatter **trajectory** of change in depressive symptoms from pregnancy to 12 months postpartum (*p = 0.048*). Conversely, at ↑ levels of **KYN/TRP ratio** and ↑ **IL-6** levels were *associated* with a ↓ in depressive scores from pregnancy to 3 (*p = 0.03*) and 12 months (*p = 0.014*) postpartum. |
| Teshigawara (2019) (N = 132) Japan | To determine whether cytokines and kynurenine metabolites can predict the development of depression in pregnancy. | Prospective ≤ 25 and ~36 weeks gestation; 1 month postpartum. | Tryptophan pathway | In the non-depressed group: **TRP, KYN, 3HK,** and **KA** were ↑ postpartum compared to pregnancy (two-way repeated ANOVA, **Trp:** $F_{group (3, 128)}$ = 1.44, p = 0.234, $F_{period (1, 128)}$ = 64.3, p < 0.0001, $F_{group x period (3, 128)}$ = 0.376, p = 0.771; **Kyn:** $F_{group (3, 128)}$ = 0.927, p = 0.430, $F_{period (1, 128)}$ = 96.4, p < 0.01, $F_{group x period (3, 128)}$ = 6.09, p < 0.01; **3HK:** $F_{group (3, 128)}$ = 0.0662, p = 0.978, $F_{period (1, 128)}$ = 6.09, p < 0.05, $F_{group x period (3, 128)}$ = 1.98, p = 0.120; **KA:** $F_{group (3, 128)}$ = 1.52, p = 0.213, $F_{period (1, 128)}$ = 2.11, p = 0.149, $F_{group x period (3, 128)}$ = 5.32, p < 0.01). In the postpartum depressed group: **KYN** and **KA** were ↑ during pregnancy, but **3HAA** during the postpartum period was ↓ than that of the non-depressed group. No differences were noted in **TRP** or its metabolites between the temporary gestational depressive group or the continuous depressive group and the non-depressive group. The ratio of **KYN** in the postpartum period compared to that during pregnancy was significantly ↓ in the postpartum depressive group compared to the non-depressive group (*one-way ANOVA, $F_{(3, 128)}$ = 5.27, p < 0.01*). In the postpartum depressive group **KYN/TRP** and **KA/KYN** ratio during pregnancy were ↑ than those in the non-depressive group. **KYN/TRP** during postpartum to that during pregnancy was significantly ↓ than the non-depressive group (*one-way ANOVA, $F_{(3, 128)}$ = 4.54, p < 0.01*). **KYN, KA,** and **KYN/TRP,** and **KA/KYN** ratio during pregnancy were ↑ and **3HAA** during postpartum was ↓ in the postpartum depressive group compared to non-depressive group. |
| ®Robertson Blackmore[2] (2016) (N = 171) US | Examine the relationship between exposure of intimate partner violence (IPV) and proinflammatory cytokine levels, a candidate mechanism accounting for poor psychiatric and obstetric outcomes, across the perinatal period | Prospective 18- and 32-weeks gestation (± 1 week); 6 weeks and 6 months postpartum (± 1 week). | Inflammatory markers (significant life events) | **KYN, KA,** and **KYN/TRP** during pregnancy was *correlated* with depression scores during the postpartum period (*Pearson's correlation:* **KYN:** $r_{(77)}$ = 0.330, p < 0.01, **KA:** $r_{(77)}$ = 0.278, p < 0.05, **KYN/TRP:** $r_{(77)}$ = 0.229, p < 0.05, **KA/KYN:** $r_{(77)}$ = 0.221, p = 0.05). There was a *negative relationship* between **3HAA** levels during postpartum period and depression scores (*Pearson's correlation: $r_{(77)}$ = -0.259, p < 0.05*). Those with a history of **IPV** had ↑ levels of **TNF-α** (*z = -2.29, p < 0.05*) compared to those with no IPV exposure. After *controlling* for participants characteristics, a greater change in the levels of **IL-6** during pregnancy compared to the postpartum period remained (*β = 0.21, p = 0.04*). This trend was different according to **IPV status.** Those who **experienced violence** had smaller changes in **IL-6** across the time points compared to those not exposed to violence (*β = -0.36, p = 0.04*). From 6 weeks to 6-month PP, those **exposed to violence** had a greater ↓ in **IL-6** compared to those without exposure (*β = 0.36, p = 0.04*). The change in **TNF-α** levels at 32 weeks' gestation to 6 weeks PP was ↑ than the change from 6 weeks to 6 months PP (*β = 1.54, p < 0.01*). |

(*Continued*)

**Table 2.** (Continued)

| Study | Objective | Study design/Timeframe | Factor | Results |
|---|---|---|---|---|
| Fasching (2012) (N = 361) *Germany* | Identify trajectories of perinatal depressive symptoms and their predictors among low-income South African women who were already at risk of depression during pregnancy. | Prospective *Third trimester, 2–3 days postpartum, 6–8 months postpartum. Unknown when blood samples were collected other than postpartum.* | Genetic polymorphisms | Haplotype block analysis showed that 10 of the 14 haplotypes of the *THP2* gene were assembled in three haplotype blocks (B1-B3). **SNPs rs6582071** and **rs11178997** (haplotype A) were also analyzed given these SNPs are known to be of functional relevance. <br><br>***Genotype-phenotype association in haplotype Block A:*** <br>The most common haplotype was **GT** (63.4% homozygous for this haplotype and 31.6% had one allele for GT). The extremely rare haplotype GA (only one carrier) was excluded. <br>The *linear mixed model* indicated an effect for **time** ($p < 0.00001$, *F-test*) as well as haplotype **GT** ($p = 0.02$, *F-test*) and the **interaction of time and haplotype GT** ($p = 0.03$, *F-test*). <br>*Pairwise comparison* demonstrated ↑ depression scores at different timepoints: 1) time point 3 for those **non-carriers of the GT** haplotype compared to those carrying **one copy of GT** at time point 3 ($p < 0.01$). At timepoints 1 and 3, those **non-carriers of the GT haplotype** showed ↑ depression scores than those carrying **two copies of the GT** ($p = 0.01$; $p = 0.01$). ↑ depression scores were found at timepoint 1 compared to timepoint 2 in all three haplotype groups (**0 GT**: $p < 0.001$, **1 GT**: $p < 0.01$, **2 GT**: $p < 0.00001$). There was an ↑ in depression scores from **timepoint 2 to timepoint 3 for non-carriers of a GT haplotype** ($p = 0.01$) and for carriers of **two copies of GT** ($p < 0.001$). <br><br>***Haplotype block B1:*** <br>**SNPs:** rs6582071, rs11178997, rs1117899; **Haplotypes:** CAT, CGA, CGT, TAA <br>Results are identical to those from haplotype block A described above. <br><br>***Haplotype block B3:*** <br>Block B3 resulted in four haplotypes (**GAA, TAA, TA, TTG**) with the most common being **TTA**. 33% of those carrying **two copies** and 51.8% carrying **one copy**. <br>*Linear mixed model:* Those carrying **two copies of TAA** (0.6%) were joined with the carriers of **one copy of TAA** (15.5%). An effect for **time** was shown ($p < 0.00001$, *F-test*) as well as the interaction between **TAA and time** ($p = 0.01$, *F-test*). Differences between the patient groups at **time 1** were seen for **TAA**, and **both genotype groups** were different between all three **time points** ($p < 0.00001$, $p < 0.00001$, $p < 0.01$). <br>*Pairwise comparison:* **Three timepoints** showed ↑ depression scores at **time 1** and **time 2** for **TAA** (*0 TAA*: $p < 0.0001$, $1 + 2$ TAA: $p < 0.0001$). At **time 2** and **3**, an ↑ in depression scores was seen in both groups ($0$ TAA: $p = 0.03$, $1 + 2$ TAA: $p = 0.02$), and depression scores were ower at **time 1** compared to **time 3** ($0$ TAA: $p < 0.01$, $1 + 2$ TAA: $p < 0.01$). <br><br>***SNPs outside of haplotype blocks:*** <br>**rs10879354** (T/T + T/C vs C/C) showed an effect for **time** ($p < 0.00001$) and **SNP** ($p = 0.04$) but not for interaction. <br>*Pairwise comparison* of the **three timepoints** showed ↑ depression scores at **time 1** compared to **time 2** ($p < 0.00001$); **time 2** compared to **time 3** indicated a depression score ↑ ($p < 0.001$); **time 3** was ↑ than **time 1** ($p < 0.01$). |
| Lin[2] (2009) (N = 200) *Taiwan* | To determine whether cytokines and kynurenine metabolites can predict the development of depression in pregnancy. | Cross sectional *Unclear for biospecimen collection. Mood disorder assessed at 36 weeks gestation, and 8 and 18 weeks postpartum* | Genetic polymorphisms | Six SNPs (**T-703G, T-473A, A90G, C2755A, C10662T, G93329A**) were noted from the *TPH2* gene. <br>Two SNPs were found in the cases (*T-473A*, $p = 0.042$; *A90G*, $p = 0.038$) that were not found in controls. <br>*Risk analysis* showed that the **"A" allele** conferred a risk (*RR = 1.73*; 95% CI: 1.59–1.88) and demonstrated a dominant gene effect (*A-allele carrier vs non-A allele carrier, AC vs CC*; $p = 0.038$). <br>A strong linkage disequilibrium in the 5' region between **SNPs -703A** and **A90G** in both groups (*D' ranged from 0.87 to 1*) and the *D'* dropped as the distance between the pairs of markers ↑ (*D' ranged from 0.50–0.76*). <br>The **GTAA haplotype**, which contains the risk **2755A allele**, was different among patients and controls (*Fisher's exact test*, $p = 0.044$); however, the significant in distribution of the **GTAA haplotypes** disappeared in a rigid permutation test ($p = 0.086$). |

® Study reported race and/or ethnicity; Author[2] = secondary analysis; US, United States; Factors investigated in relation to depression **bold**; Timeframe and/or groups investigated underlined; *Values (when provided)* = statistical values respective to analysis.

interpersonal relationships have potential to induce inflammatory responses, Ross and colleagues (2018) found romantic partner relationships low in both negative (e.g., conflict) and positive (e.g., support, intimacy) features to be associated with lower anti-inflammatory cytokines (IL-10, IL-13) and higher pro-inflammatory profile (IL-6:IL-10 ratio). Whereas Finy and colleagues (2018) found past (i.e., childhood abuse) and current adversities (i.e., lower SES) to be positively associated with elevations in inflammatory markers (i.e., CRP, IL-6).

A positive association among depression symptoms and IL-6 (involved in both immune response and inflammation) was found [46, 54, 58]. Even when controlling for pre-pregnancy body mass index (BMI), higher depression scores were positively associated with both IL-6 and TNF-α [42]. Similarly, after adjusting for demographic factors and pharmacological treatment, Achytes and colleagues (2020) found that postpartum individuals with elevated plasma levels of IL-6, IL-8 (pro-inflammatory cytokine), and TNF-α (modest) had increased odds of PPD while a decrease in IL-2 (pro-inflammatory cytokine) increased the odds of PPD. No associations with increasesd risk of PPD were found for plasma IL-10 (anti-inflammatory cytokine) or IL-1β (pro-inflammatory cytokine) were found [39]. Results from Sha and colleagues (2022) further support the aforesaid findings for IL-6, and go on to suggest a potential second-trimester biomarker panel (IL-6, TNF- α, quinolinic, and kynurenine) to predict PPD.

Though there were consistencies among findings for IL-6 and TNF- α [39, 42, 46, 50, 54, 58], some studies presented contradictory findings for certain inflammatory markers detailed above [42, 56, 58]. For instance, Sha and colleagues (2022) found IL-1β to be negatively associated with depression scores across four-time points (i.e., three trimesters, one postpartum time point) while Achtyes and collegues (2020) found no associations. Christian and colleagues (2009) found that depression scores were positively correlated with IL-2 and IL-10 rather than negatively correlated [39, 42], and Robertson-Blackmore and colleagues (2016) did not find depressive symptoms in the third trimester to be associated with IL-6 or TNF-α [39, 42, 46, 50, 54, 56, 58]. Differences in perinatal timepoints assessed and additional methodological differences between studies may explain conflicting results.

Lastly, depression was positively associated with oxidative stress during pregnancy, as measured by 8-isoprostane (considered a stable biomarker of oxidative stress) in urine, and oxidative stress mediated the relationship between prenatal depression and spontaneous preterm birth [63]. While sources of oxidative stress vary, evidence suggests the sources, in part, are related to environmental and lifestyle factors [93, 94]. Therefore, it may be meaningful to investigate factors that influence oxidative stress in the perinatal period in relation to associated health outcomes (i.e., depression, spontaneous preterm birth) to explore how such factors may be attenuated and leveraged for risk mitigation.

**Tryptophan pathway, metabolites, and neurotransmitters.** Brain TRP availability was negatively associated with plasma competitor amino acid concentrations during the postpartum period (+77% isoleucine, +55% leucine, +52% tyrosine) and the intensity of postpartum "blues" [40]. It is important to note that though we acknowledge postpartum blues as different than PPD, the difference is largely the duration of symptoms as postpartum blues is considered transient. The timepoint investigated by Bailara and colleagues (2006) was three days postpartum, meaning it is unknown if these symptoms were in fact transient or if symptoms continued beyond study participation and were later considered PPD. Therefore, for transparency, we retained the use of the term postpartum blues and decided to include these findings given the findings are consistent with those in non-perinatal populations yet is understudied in perinatal populations [24, 95].

Plasma levels of serotonin and neuropeptide Y (stimulates food intake, particularly carbohydrates) were lower in those with PPD [55]. Conversely, dopamine (role in movement, motivation, pleasure) and norepinephrine (role in flight-or-fight response) were higher in those

with PPD compared to controls. Achytes and colleagues (2020) also found that lower plasma serotonin increased the risk of PPD, whereas absolute plasma levels of TRP did not affect the risk of PPD. Though not specific to depression, Achytes and colleagues (2020) found that suicide, a distal outcome of depression and a leading cause of maternal mortality, was associated with lower levels of plasma serotonin and lower plasma serotonin increased the odds of a completed suicide attempt during pregnancy even when adjusting for EPDS scores. Though such findings require further investigation, serotonin may be significant biomarker of suicide risk in perinatal populations.

Prenatally, plasma levels of kynurenine (KYN) and kynurenic acid (KA) were significantly higher in the depressed group compared to the non-depressed group. Postpartum, higher plasma levels of KYN, KA, and KYN/TRP and KYN/KA ratios were observed in the PPD group compared to those in the non-depressed group [60]. Sha and colleagues (2022) found quinolinic acid, a potentially neurotoxic TRP metabolite that gets synthesized via the KYN pathway, to be associated with depression in the third trimester. Higher plasma levels of quinolinic acid were associated with both increased severity and risk of falling into a category of clinically significant symptoms (i.e., EPDS $\geq$13). In non-perinatal populations with depression, inflammation is suggested to play a role in the shunting of TRP down the KYN pathway and KYN has become increasingly recognized as a potential link between inflammation and depression [24, 96]. KYN has also been linked with sleep disturbances, a common depression symptom, which is also commonly experienced perinatally [96, 97]. Poor sleep has also been widely established as a risk factor for a number of chronic health conditions. For these reasons, it may be beneficial for future research to explore such interactions and the directionality of said interactions as they relate to perinatal depression onset, chronicity, and risk for comorbidities.

Conversely, Veen and colleagues (2016) found KYN to be significantly lower in patients with perinatal depression compared to non-depressed controls. Similarly, findings from Nazzari and colleagues (2020) suggest a negative association among prenatal KYN levels and depression symptoms in late pregnancy and postpartum after adjusting for maternal age. No differences were found in the plasma levels of TRP or its metabolites among perinatal depressed groups compared to non-depressed controls [60, 62]. Kimmel and colleagues (2022) found no significant associations among TRP/serotonin related metabolites or bile acids and depression. While three studies [48, 54, 62] provided conflicting results related to KYN levels, differences in the timepoints assessed, the country where the study took place, and differences in methodological decisions may explain the conflicting results as lifestyle choices and psychosocial and environmental factors are likely quite different between countries. Two of the three studies [48, 62] were also likely underpowered as one was a pilot study with a sample size of 30 and the second had a sample size of 42, with 23 being cases of PPD while the remaining were controls. Lastly, as previously discussed, sleep disturbances have been linked to the KYN pathway and depression, and inflammation is suggested to increase the shunting of TRP down the kynurenine pathway. However, five of the seven studies examining TRP did not consider inflammation as a variable in their study nor did any of the seven studies assess sleep. Inflammation and sleep disturbances are both commonly experienced perinatally which may explain why these factors have been overlooked; however, for the reasons discussed, they are important factors to consider in the context of perinatal depression.

**Genetic polymorphisms.** Catechol-O-methyltransferase (*COMT*) is a gene that provides instruction for the metabolization of catecholamine neurotransmitters (i.e., epinephrine, norepinephrine, dopamine). A common functional polymorphism studied in relation to psychiatric conditions is the *COMT* variant, *Val$^{158}$Met* (rs4680), where an amino acid change of valine [val] to methionine [met] is suggested to reduce the activity of the COMT

enzyme that metabolizes the aforesaid neurotransmitters [98–100]. This polymorphism is suggested to influence cognition and behavior in psychiatric conditions, such as depression. Though the *COMT* variant is minimally explored in perinatal depression, Comasco and colleagues (2011) found an association among the polymorphism (*COMT-Val$^{158}$Met*) and PPD symptoms at 6 weeks but not at 6 months. Additionally, genetic variation in the Monoamine oxidase A (*MAOA*) gene is suggested to contribute to depression, specifically when *MAOA* activity is high. Higher gene activity occurs when there is a polymorphism in rs1137070 where a C allele replaces a T. Higher *MAOA* activity induced by this polymorphism may result in rapid catalyzation of the neurotransmitters serotonin and norepinephrine [101]. However, a meta-analysis suggests the T variant is associated with major depression in non-pregnant populations [102]. With regard to gene-gene interactions, Comasco and colleagues (2011) found *COMT-MAOA* interactions to be significantly associated with PPD symptoms. For instance, among low *MAOA* carriers (T allele), the *Met* variant of the *COMT* gene was related to PPD symptoms; whereas the high *MAOA* variant (C allele) was related to PPD symptoms only when combined with the *Met* allele of *COMT*. In terms of gene-environment interactions, *COMT-Val$^{158}$Met* was also associated with PPD symptoms when psychiatric history and stress were present. This interaction effect may explain why studies have reported significant associations of both *MAOA* polymorphisms with depression.

Two studies explored polymorphisms of tryptophan hydroxylase 2 (*TPH2*), the rate limiting enzyme of serotonin biosynthesis, in those with perinatal depression [45, 49]. The *TPH2* gene plays a major role in the regulation of the neurotransmitter serotonin, and genetic variants of *TPH2* are suggested to play a significant role in both susceptibility to depression and response to a commonly prescribed treatment, selective serotonin reuptake inhibitors (SSRIs) [103, 104]. Lin and colleagues (2009) found that the *TPH2 C2755A* polymorphism occurred only in those with perinatal depression and/or an anxiety disorder. Further, though significance faded after Bonferroni correction, a risk analysis demonstrated that the *TPH2 C2755A* polymorphism increased risk of perinatal depression and exhibited a dominant gene effect. However, it is important to note the reported study is specific to a Han Chinese population, and the authors position this polymorphism as a potential population specific indictor of depression risk based these findings and current evidence in Han Chinese populations. Also exploring *TPH2* polymorphisms, Fasching and colleagues (2012) found the single-nucleotide polymorphism (SNP) in intron 8 (rs10879354) to be the only SNP to show consistent effects across all time points ($\geq$ 31 weeks gestation, 48–72 hours postpartum, 6–8 months postpartum) in a German population. Since *TPH2* polymorphisms influence the activity of neurotransmitters commonly associated with depression in non-perinatal and perinatal populations [103, 104], it would be beneficial for future investigations to further examine potential genetic biomarkers and their influence on relevant metabolic pathways. Progressing this area of inquiry will not only improve the odds of discovering a genetic biomarker for perinatal depression risk but may also advance our understanding of population specific biomarkers which could drastically increase precision in early detection and intervention.

**Micronutrient alterations.** A negative association among vitamin B12, cobalamine deficiency (cB12), and serotonin were observed with probable PPD [44]. Adequate amounts of B12 are suggested to be particularly important in pregnancy for both the pregnant person as well as the offspring given its role in nervous system health [105, 106]. Concurrent with folate, B12 aids in DNA synthesis as well as red blood cell production. Interestingly, dietary sources considered high in B12 (i.e., animal-based proteins) are also sources high in TRP, the precursor to serotonin [107]. In the same study [44], a positive association was found among Methylmalonic acid (MMA) (suggested marker of functional deficiency of vitamin B12),

homocysteine (hcy) (broken down by B12 and folic acid; elevated levels suggest vitamin deficiency), and 5 methyltetrahydrofolic acid (5-methyl THF) (suggested marker of a folate/methyl trap due to existing B12 deficiency) and depression symptoms. Further, elevated MMA and 5-methyl THF were found to be significant predictors of probable PPD, and MMA was suggested to be a potential mediator of PPD.

Other micronutrient alterations that were associated with prenatal depression were total n-3 polyunsaturated fatty acids (n-3 PUFA), eicosapentaenoic acid (EPA), and docosahexaenoic acid (DHA). EPA and DHA are two notably important fatty acids given they are critical for the development and function of the central nervous system in both perinatal persons and the developing fetus and have anti-inflammatory properties [108]. Chang and colleagues (2018) found that those with prenatal depression had lower levels of total n-3 PUFA, EPA, and DHA compared to those without prenatal depression. Prenatal depression duration was negatively correlated with total n-3 PUFA, EPA, and DHA. These findings indicate micronutrient deficiencies, either due to low dietary consumption and/or an existing functional deficiency, may be useful measures for detecting perinatal depression risk or early symptom onset. Though supplementation of B12 has provided conflicting results, reviews and meta-analyses of RCTs have shown the fatty acids discussed can improve depression symptoms in perinatal and non-perinatal populations and may be useful as an independent or adjuvant treatment modality depending on the individual [109–111]. This is particularly important to note for perinatal populations as pharmacological interventions are not highly desired in pregnant or breastfeeding persons due to concerns for potential implications on the developing offspring [112, 113]. Thus, though future research is needed, micronutrient supplementation may be desirable option for those at risk or those exhibiting mild depressive symptoms during pregnancy to serve as form of risk mitigation and indirect health promotion strategy for the developing offspring.

### Neurological factors

Brain serotonin-1A (5HT1A) receptor binding potential, as measured by positron emission tomography (PET), suggested a 20–28% reduction in postsynaptic 5HT1A receptor binding in those experiencing PPD [53]. The most significant reductions were found to be in the anterior cingulate (related functions—emotional expression, attention, mood regulation) and mesiotemporal cortices which includes the amygdala (input and processing of emotion) and hippocampus (episodic memory). Likely due to methods of data acquisition and the unknown risks for perinatal individuals and their offspring, investigations into neuroanatomical features and their respective roles in perinatal specific depression are sparse. While there are some conflicting findings, the present findings are consistent with some literature on depression and/or anxiety in non-perinatal populations [114, 115].

### Behavioral determinants

Six studies [39, 44, 47, 51, 52, 61] investigated behavioral determinants of perinatal depression. Dietary intake was investigated in three studies [44, 51, 52] whereas suicide (i.e., attempts, ideation, risk) was explored in two [39, 47]. Physical activity [61] and functional impairment [47] were each explored in one. A summary of findings with statistical values for each of the six studies can be found in **Table 3**.

Though specific to the third quartile (ascending quartiles per glycemic index/load), Murakami and colleagues (2008) found higher dietary glycemic index (GI) decreased the risk of PPD, while no associations were found between PPD and dietary glycemic load (GL). Lower milk, meat, and egg consumption during the postpartum period was associated with probable

**Table 3. Summary of findings—behavioral determinants.**

| PREGNANCY (Behavioral) | | | | |
|---|---|---|---|---|
| **Study** | **Purpose/Aims** | **Design** | **Determinants** (*from other domain*) | **Summary of findings** (*values*) |
| | | **Timepoints** | | |
| Miyake[2] (2022) (N = 1744) *Japan* | Examine the association between tryptophan intake and depressive symptoms during pregnancy. | Cross sectional *5–39 weeks gestation* | Dietary intake | **Tryptophan intake** was *positively associated* with being **unemployed** (*p = 0.0001*), **household income** (*p = 0.002*), **education** (*p = 0.01*), and intake levels of **saturated fatty acids** (*p ≤ 0.0001*), **eicosapentaenoic acid plus docosahexaenoic acid** (*p ≤ 0.0001*), **calcium** (*p ≤ 0.0001*), **vitamin D** (*p ≤ 0.0001*), **isoflavones** (*p ≤ 0.0001*), **fish** (*p ≤ 0.0001*) and *negatively associated* with having ever **smoked** (*p = 0.0006*) and **cereal intake** (*p ≤ 0.0001*). **Age** was *negatively associated* to the prevalence of depressive symptoms during <u>pregnancy</u> in a crude analysis (*p = 0.02*). |
| | | | | Compared with **tryptophan intake** in the <u>lowest quartile</u>, **tryptophan intake** in the <u>highest quartile</u> was related to a ↓ prevalence of depressive symptoms during <u>pregnancy</u>. The *inverse exposure–response* relationship was also significant in the *crude analysis*. |
| | | | | ↑ **tryptophan intake** was independently <u>negatively associated</u> with the prevalence of depressive symptoms during <u>pregnancy</u>: the *adjusted PRs* (95% CIs) for depressive symptoms during <u>pregnancy</u> in all <u>four quartiles</u> of **tryptophan intake** (*Crude PR (95% CI)*, 1.00; 0.95 (0.74–1.22); 0.87 (0.67 −1.12); 0.57 (0.42−0.76), p = 0.0001)); (Adjusted PR (95% CI), 1.00; 0.99 (0.76 −1.28); 0.94 (0.71−1.25); 0.64 (0.44−0.93), p = 0.04)). |
| | | | | These results were not changed when controlling for dietary factors. |
| POSTPARTUM (Behavioral) | | | | |
| Dhiman[2] (2021) (N = 660) *India* | Explore the association between vitamin B12 and probable PPD in South Indian population. | Cross sectional 6 weeks postpartum | Dietary intake (micronutrient alterations) | They also reported ↓ **milk** (*p < 0.001*), **meat** (*p = 0.012*), and *egg* (*p = 0.002*) **intake.** |
| Achtyes (2020) (N = 130) *US* | Investigate whether a pro-inflammatory status in plasma, together with changes in the kynurenine pathway activity, is associated with the development of severe depression and suicidal behavior in the post-partum. | Case-control *6–12 weeks postpartum; pregnancy timepoint(s) not specified for measuring suicidality* | Suicide (inflammatory markers; tryptophan pathway) | ↓ **Serotonin** significantly ↑ the odds of a **completed suicide attempt** during <u>pregnancy</u>. This association remained even when adjusting for EPDS score (69.9% increased odds per nM decrease in serotonin, 95% CI: 2.1% - 182.8%, p = 0.044). **Quinolinic acid** levels significantly impacted the odds of **suicidal ideation** but did not affect suicidal behavior (OR: 0.1, CI 95% [0.02–0.51], p = 0.006; OR: 0.1, CI 95% [0.09–0.36], p = 0.11). |
| PERINATAL (Behavioral) | | | | |

**Table 3.** (Continued)

| | | | | |
|---|---|---|---|---|
| Garman[2] (2019) (N = 384) *South Africa* | Identify trajectories of perinatal depressive symptoms and their predictors among low-income South African women who were already at risk of depression during pregnancy. | Prospective *First or second trimester, 8 months gestation, and 3 and 12 months postpartum.* | Suicide; functional impairment (Significant life events; social support; food insecurity) | *Odds* of belonging to the <u>prenatal and postpartum depression</u> class were ↑ among those who reported **greater functional impairment** *(OR = 1.03, 95% CI: 1.02–1.06; p = 0.002)*, **heavy drinking during pregnancy** *(OR = 2.12, 95% CI: 0.03–4.37; p = 0.042)*, had **current** *(OR = 2.77, 95% CI: 1-32-5.80; p = 0.007)* or **lifetime diagnosis of depression** *(OR = 2.85, 95% CI: 1.38–5.87; p = 0.004)*, and **high risk of suicide** *(OR = 2.58, 95% CI: 1.19–5.61; p = 0.017).* |
| Vargas-Terrones (2017) (N = 124) *Spain* | Analyze trajectories of serotonin and tryptophan-related metabolites, bile acid metabolites, and microbial composition related to psychiatric history and current symptoms across the perinatal period. | Randomized control trial *<16 weeks gestation, intervention 12–16 gestation through 38–40 weeks gestation, follow up—6 weeks postpartum* | Physical activity | The percentage of depressed participants was ↓ in the **intervention group** compared to the control group at <u>week 38</u> *(18.6% vs. 35.6%)* $(\chi^2 = 4.190; p = 0.041)$ and at <u>6 weeks postpartum</u> *(14.5% vs 29.8%)* $(\chi^2 = 3.985; p = 0.046).$ |
| | | | | Significant differences were noted in the *multiple imputation analysis* at <u>38 weeks</u> *(18.6% vs. 34.4%)* $(\chi^2 = 4.085; p = 0.049).$ |
| | | | | A **treatment effect** was found in the per-protocol $(F_{2,\ 220} = 3.798; p = 0.024)$ and in the *simple imputation* $(F_{2,244} = 3.351; p = 0.037)$ analyses. Differences were also found in the *group-time interaction* between gestational weeks 12–16 <u>(baseline)</u> and <u>6 weeks postpartum</u> *(p = 0.014)* in the per-protocol analysis. |
| | | | | Differences were found in the **group-time interaction** between depression scores at <u>baseline</u> and gestational <u>week 38</u> *(p = 0.046)*, and between <u>baseline</u> and <u>6 weeks postpartum</u> *(p = 0.025)*, with a ↓ depression score in the <u>intervention group</u> than in the <u>control group</u>. |
| | | | | The participants considered to have **excessive gestational weight gain**, the <u>control group</u> had a ↑ percentage of depression at <u>week 38</u> $(\chi^2 = 9.489; p = 0.002)$ and at <u>6 weeks postpartum</u> $(\chi^2 = 5.202; p = 0.023).$ |
| | | | | The percentage of depressed women was ↓ in the **intervention group** compared to the <u>control group</u> at <u>week 38</u> for those with **pre-pregnancy normal-weight BMI** $(\chi^2 = 4.688; p = 0.030).$ |
| Murakami[2] (2008) (N = 865) *Japan* | To examine the association between dietary GI and glycemic load (GL) and postpartum depression. | Prospective *Unspecified pregnancy timepoint and 2–9 months postpartum* | Dietary intake | Compared with **dietary glycemic index (GI)** in the <u>first quartile</u>, **dietary GI** in the <u>third quartile</u>, but not the <u>fourth</u> was *associated* with ↓ risk of <u>PP</u> depression. *Multivariate ORs* (95% Cis) for PP depression for each of the <u>four quartiles</u> were: *1.00 (reference), 0.68 (0.39–1.17), 0.56 (0.32–0.995, p = 0.048), and 0.72 (0.41–1.26), respectively (p for trend = 0.18).* |

®Study reported race/ethnicity; Author[2] = secondary analysis; US, United States; Factors investigated in relation to depression **bold;** Timeframe and/or groups investigated <u>underlined</u>; *Values (when provided)* = statistical values respective to analysis.

PPD [44]. Further, even after adjusting for potential dietary and non-dietary confounding factors, higher tryptophan intake was independently negatively associated with depressive symptoms in pregnancy [51]. TRP, the precursor to the neurotransmitter serotonin that is commonly associated with depression, is an essential amino acid. Essential amino acids, such as TRP, are only made available through dietary intake as they are not independently produced by the body [24]. Though Dhiman and colleagues (2021) did not specifically examine TRP, animal and plant-based proteins (e.g., milk, meat, eggs, spirulina, nuts and seeds) are among some of the highest sources of TRP. Thus, together these findings indicate lower dietary consumption of TRP in the perinatal period may contribute to increased risk of depression onset which is consistent with findings from animal and human studies in non-perinatal populations [24, 28, 116, 117]. Additionally, though not a variable noted in any of the included studies, nausea and vomiting due to "morning sickness" or hyperemesis gravidarum (severe type of morning sickness) occurs in roughly 70% of pregnancies [118]. Thus, these variables may be particularly important to consider in investigations of TRP metabolism perinatally as these variables are likely to increase risk for depletion of essential nutrients vital for maternal and fetal health.

Two studies independently demonstrated alcohol consumption [47] during pregnancy or a high risk of suicide (i.e., current and past attempts and ideation) [39, 47] increased the odds of being classified into the perinatal depression group. Those endorsing higher functional impairment had increased odds of being classified in the perinatal depression group [47] whereas a single randomized control trial (RCT) [61] demonstrated a prenatal physical exercise program to decrease the risk of PPD. The RCT consisted of 60-minute sessions three times per week starting at 12–16 weeks gestation and found the percentage of people reporting depression was lower in the intervention group than in the control group at both 38 weeks gestation and 6 weeks' postpartum. The findings related to substance abuse, history of suicide attempts or current suicidal ideation, and physical activity are consistent with existing literature [7, 20, 119, 120]. Conversely, functional impairment is a less commonly studied factor [121]. Functional impairment is a marked feature of clinical depression yet is not routinely assessed, if at all, during the perinatal period. Exploring the implications of functional impairment in the perinatal period may be particularly useful to clinically monitor for declines from baseline functioning for those with and without pre-existing disabilities or functional impairments. Further research in this area may help advance detection strategies for life-stage specific onset or exacerbations of pre-existing functional impairment that may not otherwise be visible to clinicians and provide evidence for identifying individuals in need of increased support. While we understand these behavioral factors may increase the risk of PPD and behavioral interventions targeting such factors may aid in mitigating risk, it is important to consider who is disproportionately impacted by perinatal depression and the broader contextual factors that serve as potential barriers and are beyond the immediate control of the individual (e.g., social determinants of health).

## Social and environmental determinants

A total of seven studies [42, 43, 46, 47, 56, 57, 59] investigated social and environmental determinants of perinatal depression. Five studies investigated significant life events (e.g., trauma, intimate partner violence, history of childhood abuse) [43, 46, 47, 56, 59] and social support [42, 43, 46, 47, 57]. Perceived stress was investigated in two studies [42, 43] and unhappiness with pregnancy [43] and food insecurity [47] were investigated in one study. A summary of findings with statistical values for each of the seven studies can be found in **Table 4**.

Two commonly suggested risk factors of perinatal depression, psychiatric history [43] and significant life events [43, 46, 47, 56, 59], were positively associated with perinatal depression

**Table 4. Summary of findings–social and environmental determinants.**

| Study | Purpose/Aims | Design / Timepoints | Determinants (from other domain) | Summary of findings (values) |
|---|---|---|---|---|
| ®Finy[2] (2018) (N = 214) US | Examine the association between childhood abuse, low socioeconomic status (SES) and inflammatory markers during pregnancy | Cross sectional ≤ 31 weeks gestation or ~30 days after flu vaccination | Significant life events; social support (inflammatory markers) | **Childhood abuse history** was *positively associated* with **CRP** and **IL-6**. |
| | | | | **Current SES** and **CRP** and **IL-6** were *negatively associated (p's < 0.01).* |
| ®Ross (2017) (N = 90) US | Examine the association between pregnant women's close relationships and cytokine profiles in the third trimester. | Prospective 22-26- and 32-36-weeks gestation | Social support (inflammatory markers) | **Romantic partner (RP) relationships** with **positive features (i.e., support/closeness)** were associated with ↓ levels of **inflammatory cytokines; RP relationships low** in both **positive and negative features (indifferent)** were *associated* with **cytokine profiles** indicating ↑ **inflammation**. |
| | | | | **Positive RP** relationship was negatively associated with **IL6:IL10 ratio**. Further, when **positive RP** features were ↑ and there were ↓ **RP negative** features, the estimated **IL6:IL10 ratios** were lowest indicating a potential buffering or protective effect of **positive RP relationships**. |
| | | | | **Positive** and **negative RP relationships** were *associated* with **IL10** levels *(b(SE) = 0.031 (0.009), p = 0.001; b(SE) = 0.017 (0.007), p = 0.017).* |
| | | | | **Positive** and **negative RP relationships** were *associated* with **IFNy** levels *(b(SE) = 0.131 (0.041), p = 0.002; b(SE) = 0.095 (0.032), p = 0.004)* |
| | | | | Neither **positive** and **negative RP relationships** were *associated* with **IL13, IL8, IL6,** and **TNF-α** levels. |
| | | | | ↑ **positive RP relationship** was *associated* with ↓ depressed mood *(r = -0.35, p = 0.001)* and **perceived stress** *(r = -0.41, p < 0.001)* whereas ↑ **negative RP relationship** was *associated* with ↑ depressed mood *(r = 0.51, p < 0.001)*, **perceived stress** *(r = 0.53, p < 0.001)*, and **pregnancy distress** *(r = 0.29, = 0.005).* |
| ®Christian (2009) (N = 60) US | Examine associations among perceived stress, current depressive symptoms, and serum inflammatory markers among pregnant women from primarily lower socioeconomic backgrounds. | Cross-sectional First and second trimester | Social support; stress (inflammatory markers) | Depressive symptoms were *positively correlated* with **perceived stress** *(r = 0.050, p < 0.01).* |
| | | | | Those classified as **unhappy about their pregnancies** had ↑ depressive symptoms compared to those who were happy about their pregnancy *(mean CES-D = 22, SD = 10; mean CES-D = 16, SD = 10, p = 0.04).* |
| | | | | Those reporting ↓ **social support** had ↑ depressive symptoms *(p < 0.05)*, and those with ↑ frequent **hostile and insensitive social interactions** also had ↑ depressive symptoms *(p < 0.01).* |
| | | | | After *controlling* for **social support**, **hostile and insensitive social interactions** remained *associated* with depressive symptoms *(β = 0.17, r(1, 59) = 1.25, p = 0.21).* |
| **PREGNANCY (Social and Environmental)** |||||

| POSTPARTUM (Social and environmental) |
|---|

| Comasco (2011) (N = 272) Sweden | Examine whether genetic variations in the monoaminergic neurotransmitter system, together with environmental stressors, contribute to the development of PPD symptoms | Case control 6 weeks and 6 months postpartum | Significant life events; social support, stress; unhappiness with pregnancy (genetic polymorphisms) | Previous **psychiatric contact**, **significant life events**, and **maternity stressors** were *associated* with PPD symptoms. |
|---|---|---|---|---|
| | | | | *COMT*Val[158]Met was *associated* with PPD symptoms in the presence of **previous psychiatric contact** and **maternity stressors**, while **MAOA-uVNTR** was *associated* with PPD symptoms only in the presence of maternity stressors. |
| | | | | The *logistic regression analysis* demonstrated an *association* among PPD symptoms and **COMTVal[158]Met, previous psychiatric contact**, and **maternity stressors**. The *model explained 30% variance*. After stratifying for previous psychiatric contact, the gene-environment interaction model indicated those with **previous psychiatric contact** had a *main effect* of **COMT-Val[158]Met** and **5HTT-LPR** with an *explained variance of 40%*. |

| PERINATAL (Social and environmental) |
|---|

(*Continued*)

**Table 4.** (Continued)

| | | | | |
|---|---|---|---|---|
| ®Tebeka[2] (2021) (N = 3,252) *France* | Assess the relationship between childhood trauma (CT) and perinatal depression, considering types of CT | Case control *2–5 days postpartum, follow up at 8 weeks and 1 year postpartum (depression in pregnancy was assessed postpartum)* | Significant life events | Those reporting **childhood trauma (CT)** were ↑ likely to be < **26 years old** *(8.1% vs. 4.5%; OR = 1.8; 95% CI: 1.2–2.6)* > **39 years old** *(11% vs. 7%; OR = 1.9; 95% CI: 1.2–2.9)*, **single** *(6.7% vs. 2.7%; OR = 2.6; 95% CI: 1.5–4.2)*, **have a lower level of education** *(18.1% vs. 6.8%; OR = 3.0; 95% CI: 1.8–3.6)*, and ↑ likely to have been **unemployed** *(14.1% vs. 6.1%; OR = 2.5; 95% CI: 1.8–3.6)*. |
| | | | | Those with **CT** had a ↑ risk of either depression, anxiety, or suicide attempts compared those without *(61.6% vs. 40.8%; OR = 2.3; 95% CI: 1.8–2.9)*, and a personal history of depression, anxiety, or suicide attempts were ↑ frequent in those with **CT** (depression: *OR = 2.2; 95% CI: 1.7–2.7;* anxiety: *OR = 2.3; 95% CI: 1.7–3.0;* suicide attempt: *OR = 5.4; 95% CI: 3.5–8.4)* |
| | | | | Depression was ↑ common in those with a **CT** regardless of **type of CT**, and the difference was significant for **emotional, physical**, and **sexual abuse** as well as **emotional neglect** *(p < 0.05 for each)*. The **types of CT** demonstrated specific *associations* with different timing of depression onset. **Emotional neglect** was *associated* with depression during pregnancy *(aOR = 2.1; 95% CI: 1.2–3.8, p = 0.012)*; **sexual abuse** with both early and late onset PPD *(aOR = 2.3; 95% CI: 1.2–4.6; aOR = 2.4; 95% CI: 1.2–4.9, respectively)*; **emotional abuse** was *associated* only with late PPD *(aOR = 2.7; 95% CI: 1.4–5.1)*. |
| | | | | A *dose effect* was present between **CT types** and risk of depression. When **1 type of CT** was present there was a ↑ risk of depression *(aOR = 1.6; 95% CI: 1.1–2.3, p = 0.015)*, whereas, when **2+ types of CT** were present the risk further ↑ *(aOR = 2.1; 95% CI: 1.3–3.3)* even after adjusting for history of depression and sociodemographic covariates. |
| Garman[2] (2019) (N = 384) *South Africa* | Identify trajectories of perinatal depressive symptoms and their predictors among low-income South African women who were already at risk of depression during pregnancy. | Prospective *First or second trimester, 8 months gestation, and 3 and 12 months postpartum* | Significant life events; social support; food insecurity (suicide; functional impairment) | **Food insecurity** *predicted classification* of either prenatal only depression or prenatal and postpartum depression. The *odds* of being classified in the prenatal and postpartum depression trajectory was 2.5 greater *(95% CI: 1.21–5.15; p = 0.013)* among participants who reported being **severely food insecure**. |
| | | | | Overall levels of **social support** at baseline ↓ the *odds* of belonging to the prenatal and postpartum depression class *(OR = 0.97, 95% CI: 0.95–0.99; p = 0.011)*. When looking at specific **types of support**, only a ↑ level of **family support** *(OR = 0.91, 95% CI: 0.86–0.96; p = 0.001)* or ↑ level of **support** from a **significant other** *(OR = 0.94, 95% CI: 0.88–1.00; p = 0.046)* ↓ the *odds* of being classified into the prenatal and postpartum depression class. |
| | | | | Those who reported **IPV** at baseline were 2.8 times ↑ likely *(95% CI: 1.23–6.52; p = 0.014)* to belong to the prenatal and postpartum depression class. |
| | | | | *Odds* of belonging to the prenatal and postpartum depression class were ↑ among those who reported **greater functional impairment** *(OR = 1.03, 95% CI: 1.02–1.06; p = 0.002)*, **heavy drinking during pregnancy** *(OR = 2.12, 95% CI: 0.03–4.37; p = 0.042)*, had **current** *(OR = 2.77, 95% CI: 1-32-5.80; p = 0.007)* or **lifetime diagnosis of depression** *(OR = 2.85, 95% CI: 1.38–5.87; p = 0.004)*, and **high risk of suicide** *(OR = 2.58, 95% CI: 1.19–5.61; p = 0.017)*. |
| ®Robertson Blackmore[2] (2016) (N = 171) *US* | Examine the relationship between exposure of intimate partner violence (IPV) and proinflammatory cytokine levels, a candidate mechanism accounting for poor psychiatric and obstetric outcomes, across the perinatal period | Prospective *18- and 32-weeks gestation (± 1 week); 6 weeks and 6 months postpartum (± 1 week).* | Significant life events (inflammatory markers) | **Lifetime exposure to IPV** was *associated* with a range of psychiatric conditions, including generalized anxiety disorder, post-traumatic stress disorder, and depression. Further, **IPV** was *associated* with experiencing depression during both pregnancy and postpartum. |
| | | | | Those with a history of **IPV** had ↑ levels of **TNF-α** *(z = -2.29, p < 0.05)* compared to those with no IPV exposure. |
| | | | | After *controlling* for participants characteristics, a greater change in the levels of **IL-6** during pregnancy compared to the postpartum period remained *(β = 0.21, p = 0.04)*. This trend was different according to **IPV status**. Those who **experienced violence** had smaller changes in **IL-6** across the time points compared to those not exposed to violence *(β = -0.36, p = 0.04)*. From 6 weeks to 6-month PP, those **exposed to violence** had a greater ↓ in **IL-6** compared to those without exposure *(β = 0.36, p = 0.04)*. |

®Study reported race/ethnicity; Author[2] = secondary analysis; US, United States; Factors investigated in relation to depression **bold;** Timeframe and/or groups investigated underlined; *Values (when provided)* = statistical values respective to analysis.

[17, 21]. Even after adjusting for sociodemographic factors, personal history of depression, and timing of depression onset, those reporting a history of childhood trauma were at higher risk of PPD, anxiety, and suicide attempts than those without [59]. A dose effect was present between the number of childhood trauma types and risk of PPD. Robertson-Blackmore and colleagues (2016) examined lifetime exposure to intimate partner violence and found lifetime intimate partner violence to increase the likelihood of experiencing perinatal depression. Those currently endorsing frequent hostile and insensitive social interactions experienced an increase in prenatal depressive symptoms [42]. Further, there was a negative relationship among social support and depression symptoms, and low social support served as a significant predictor of perinatal depressive symptoms [42, 47, 57]. Relatedly, higher negative qualities in one's interpersonal relationships were associated with greater depressed mood, perceived stress, and pregnancy distress [57]. Collectively, these findings indicate the level of past exposure, type of exposure, and current appraisal of interpersonal relationships may moderate one's level of risk for PPD.

Consistent with current evidence, perceived stress was positively correlated with perinatal depression [42, 43, 46, 57]. Being unhappy about one's pregnancy was also positively correlated with perinatal depression [42]. Such findings are particularly important to note for US based research given the current political and social climate related to child-bearing age person's rights and abortion access. Irrespective of one's personal views on the matter, the recent changes in federal and state level regulations are likely to increase rates of perinatal depression and subsequently result in a surge of negative health outcomes in both perinatal persons and the offspring. Fox and Brod (2021) investigated the cost of perinatal complications in the US for all 2019 births from conception to age 5 and found such complications to result in $32.2 billion in societal costs (i.e., healthcare expenses, loss of productivity, social support services) [122]. It was also suggested these estimates likely underrepresent the totality of the financial burden. This analysis was conducted prior to federal and state level changes on abortion access and a global pandemic. Despite spending more on healthcare than any other developed country, the US and its healthcare system have yet to gain control over the rising maternal mortality and morbidity rates. Meaning the US is likely not prepared to manage a surge in perinatal health complications, especially so soon after a global pandemic. Therefore, advancements in maternal mental health care are vital for individual and systemic health. Future investigations to further examine unhappiness with pregnancy as a potential risk factor as well as diligent monitoring of trends in incidence since the change in regulations are necessary to generate evidence for increased resources and support.

One study found food insecurity to predict perinatal depression in two groups (i.e., prenatal only depression, prenatal and postpartum depression) and the odds of experiencing depression both prenatally and in the postpartum was 2.5 greater in the presence of food insecurity [47]. These findings are supported by existing evidence; however, findings have been mixed in terms of the directionality of the relationship between food insecurity and PPD [123–125]. For instance, findings from a national dataset indicated that those experiencing food insecurity were 3.39 times more likely to report depression symptoms compared to those who were food secure [125]. Conversely, other studies have found that mothers experiencing depression may have increased experiences with socio-economic stressors, such as, food insecurity [123, 124]. Though these findings suggest a relationship among food insecurity and PPD, further research is needed to clarify the strength and direction of this relationship.

### Prospective interactions

The specific factors explored across the four domains were highly variable. Therefore, this review does not claim to present an exhaustive description of all potential interactions that can

be interpreted from findings of the aggregated factors. Due to patterns in which interactions with TRP and its metabolites emerged, we specifically chose to focus on these interactions as they may suggest a potential role in perinatal depression risk and onset.

Although minimally explored in perinatal populations, disruption in serotonin or the serotonergic system is widely considered to contribute to depression onset, maintenance, and response to treatment (i.e., SSRIs) in non-perinatal populations [104, 126, 127]. Some evidence suggests the disruption of the serotonergic system is more prominent in biologically born females compared to males, and that the dysregulation of serotonin may partly explain why biologically born females experience depression at two times the rate of biologically born males [128, 129]. Given TRP is the precursor to serotonin, brain TRP availability is vital for adequate production of the neurotransmitter serotonin. Other essential amino acids (e.g., isoleucine, leucine, phenylalanine) compete with TRP to cross the BBB and are the precursors to several other neurotransmitters (e.g., dopamine, norepinephrine) that are implicated in psychiatric conditions [95, 130]. Consistent with current evidence indicating TRP competes with other essential amino acids to cross the BBB, Bailara and colleagues (2016) found a negative association among brain TRP availability and competitor amino acid concentrations, notably one of which was tyrosine, a precursor of dopamine and norepinephrine. Since essential amino acids (i.e., TRP, isoleucine, leucine, phenylalanine) are not independently produced by the body and depend on dietary intake for availability, dietary habits, food accessibility, and other factors that may influence changes in metabolic activity (e.g., genetic polymorphisms, morning sickness, breastfeeding, comorbid conditions) are particularly important to consider in this area of inquiry. A majority of the studies in this review examining TRP and its metabolites did not concurrently examine or control for dietary habits, micronutrients, and/or food accessibility; however, a majority of the studies that did examine such factors independent of TRP found associations with PPD [41, 44, 47, 52].

The mechanisms underlying the increased uptake of TRP in the brain are not fully understood, but some evidence suggests higher dietary carbohydrate intake can promote the uptake of TRP in the brain resulting in increased serotonin [130, 131]. Interestingly, Rihua and colleagues (2018) found plasma levels of serotonin and neuropeptide Y (stimulates food intake, particularly carbohydrates) [132, 133] to both be lower in those with PPD. Achtyes and colleagues (2020) also found lower levels of plasma serotonin to increase risk of PPD but did not denote increased risk of PPD related to plasma TRP. However, they did note an elevated KYN/serotonin ratio was associated with an increased risk of PPD. TRP degradation into KYN is suggested to increase in response to immune and inflammatory activation in non-perinatal populations experiencing depression [24, 28, 83, 117]. Since pregnancy naturally induces immune and inflammatory responses in the pregnant individual to accommodate the developing fetus, TRP degradation into KYN may occur more often during this life-stage and increase one's risk for depression. Veen and colleagues (2016) did not explore the aforesaid in the pregnancy period, but they did find this to be the case in the physiological postpartum period.

Though the life-stage itself induces a unique immune and inflammatory response, additional factors throughout the perinatal period, such as the social, environmental, and behavioral factors discussed in this review (e.g., stress, social support, intimate partner violence) may further promote immune and inflammatory responses and increased TRP degradation down the KYN pathway predisposing one to depression onset. Further, sleep disturbances are common perinatally and are often attributed to "normal" pregnancy and postpartum symptoms, but sleep disturbances also happen to be a common symptom of depression and/or anxiety in non-perinatal populations. Though sleep disturbances are linked to the TRP and KYN pathways, they are not considered in any study in this review yet may serve as a moderating factor that perpetuates a negative feedback loop which contributes to chronicity or a risk for

symptom relapse, notably in the postpartum period due to poor quality of sleep as a result of child rearing responsibilities. Meaning, certain biochemical pathways may account for specific depression symptoms and indicate subtypes of perinatal depression that can be leveraged to increase precision in detection and intervention.

In the context of PPD, these findings may indicate when dietary intake of tryptophan or tryptophan uptake promoting foods are limited amid immune and inflammatory responses, competitor amino acids are being prioritized for transport across the BBB and/or TRP may be shunted towards the KYN pathway. Moreover, both pathways may result in decreased production of the neurotransmitter serotonin and explain the risk for and onset of PPD, and the level of risk would be further increased for those with a genetic predisposition (i.e., genetic polymorphisms) or those experiencing the other biopsychosocial or behavioral factors discussed in this review. Thus, future investigations are needed to further explore these prospective interactions as these interactions may serve as significant risk factors of PPD that can be detected and intervened upon during pregnancy.

## Limitations

Though this review offers prospective interactions among biological and external factors that, with further research, may provide a comprehensive understanding of perinatal depression etiology and risk, this review is not without limitations. The review is limited by its exclusion of articles not available in the English language, and the search strategy (i.e., specifying biological determinants) may have excluded relevant studies. Further, all findings from this review should be interpreted with caution given the variability across studies in regard to sample size, instruments selected to measure depression, cut-off scores for depression measures, study timepoints, and biospecimen type and methods of collection.

## Conclusion

The factors discussed in this review have been independently indicated as probable determinants of PPD risk and onset. However, what is not evident in independent investigations but is demonstrated in this review is that various interactions among diverse determinants and TRP metabolism may provide a deeper understanding of what contributes to the pathophysiology of perinatal depression or perinatal depression risk. Future investigations are needed to address methodological issues in maternal mental health science and care, and to explore prospective interactions as these interactions hold potential to evolve as a PPD risk phenotype (observable characteristics). Such a phenotype can serve as a robust foundation for the development of clinically efficient yet meaningful mechanisms for risk detection and inform patient centered risk mitigation strategies. Further, the present review establishes the value of integrative approaches in the investigation of perinatal depression and suggests the application of team science principles (e.g., collaboration, diverse expertise) may be particularly useful to this area of inquiry to expedite the discovery of clinically relevant findings and strengthen scientific methods.

## Supporting information

**S1 Checklist. PRISMA 2020 checklist.**
(PDF)

**S1 Table. Narrative descriptions of considerations in risk of bias decisions.** ®Study reported race and/or ethnicity. Author[2] indicates the study was a secondary analysis.
(DOCX)

## Acknowledgments

Thank you to all the authors and participants for your contributions in progressing maternal mental health science and care by either conducting or participating in the studies discussed in this review.

## Author Contributions

**Conceptualization:** Kayla D. Longoria.

**Data curation:** Kayla D. Longoria, Tien C. Nguyen, Oscar Franco-Rocha, Sarina R. Garcia, Kimberly A. Lewis, Michelle L. Wright.

**Formal analysis:** Kayla D. Longoria.

**Investigation:** Kayla D. Longoria.

**Methodology:** Kayla D. Longoria.

**Project administration:** Kayla D. Longoria.

**Resources:** Kayla D. Longoria.

**Supervision:** Michelle L. Wright.

**Validation:** Kayla D. Longoria, Tien C. Nguyen.

**Visualization:** Kayla D. Longoria.

**Writing – original draft:** Kayla D. Longoria.

**Writing – review & editing:** Tien C. Nguyen, Oscar Franco-Rocha, Sarina R. Garcia, Kimberly A. Lewis, Sreya Gandra, Frances Cates, Michelle L. Wright.

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
