## [Decision Letter · Decision Letter 0]

8 Sep 2023

PONE-D-23-17835A sum of its parts: A systematic review evaluating biopsychosocial and behavioral determinants of perinatal depressionPLOS ONE

Dear Dr. Longoria ,

Thank you for submitting your manuscript to PLOS ONE. After careful consideration, we feel that it has merit but does not fully meet PLOS ONE’s publication criteria as it currently stands. Therefore, we invite you to submit a revised version of the manuscript that addresses the points raised during the review process.

We look forward to receiving your revised manuscript.

Kind regards,

Xiaoqin Liu

Academic Editor

PLOS ONE

Reviewers' comments:

Reviewer's Responses to Questions

**Comments to the Author**

1. Is the manuscript technically sound, and do the data support the conclusions?

Reviewer #1: Yes

Reviewer #2: Partly

2. Has the statistical analysis been performed appropriately and rigorously? 

Reviewer #1: Yes

Reviewer #2: I Don't Know

3. Have the authors made all data underlying the findings in their manuscript fully available?

Reviewer #1: Yes

Reviewer #2: Yes

4. Is the manuscript presented in an intelligible fashion and written in standard English?

Reviewer #1: Yes

Reviewer #2: No

5. Review Comments to the Author

Reviewer #1: It was an honor and a pleasure to review the manuscript. Perinatal depression has been a popular topic of interest for a very long time. This review titled “A sum of its parts: A systematic review evaluating biopsychosocial and behavioral determinants of perinatal depression” takes an integrated approach to systematically assess the determinants of four domains of perinatal depression (i.e. biological, behavioral, environmental, social), as well as assessing the potential interactions between determinants. Overall, the article is well organized. However, some issues still need to be improved:

Introduction:

1. The authors stated that “A majority of research on the etiology of perinatal depression has attempted to dissect it into two broad camps (i.e., internal factors, external factors) ” However, the authors repeatedly used the word "biological factors" later in the text rather than internal factors. If both refer to the same meaning, it is recommended that the authors use the same word to make it easier for the reader to understand, or give examples of internal factors like external factors.

2. The authors stated that “In an era of team science, integrative approaches to investigation are not only feasible but desirable to address some of the world’s most complex health problems, like perinatal depression.” Please add references to support this statement.

3. In the Introduction, the authors do not mention whether any other studies have addressed similar issues. Therefore, the authors should mention whether there are similar studies currently and clarify what is added to this study compared to previous research. A clearer illustration of contribution or innovation should be further provided in the introduction.

Methods:

1. The authors stated that “Articles from any date were included if they focused on a timeframe within the perinatal period (i.e., conception-12 months postpartum), had participants that were 18 years or older, were available in the English language, investigated factors that belonged to at least one of the four domains (i.e., biological, behavioral, environmental, social), and had an outcome of depression or depression symptoms.” The literature search was limited to the English language, so the study might have overlooked similar studies published in other languages, which might be a limitation of this study.

2. The author should provide the advantages of CASP and AXIS compared to other quality appraisal tools.

Results and discussion:

1. The authors stated that “One article was excluded [32] during quality appraisal screening due to methodological concerns making the total articles included 25.” The author should specify the reasons for exclusion.

2. The sample sizes of the study ranged from 16 to 3,252. The conclusions drawn from the review may be affected by the low sample sizes of some studies.

3. The authors stated that “A total of four [33, 42–44] of the 25 studies discuss conducting a prior power analysis to calculate the needed sample size with half of those being US based studies [37, 42].” Please check if Ref. [33] is correctly cited.

4. The authors stated that “Given every person develops within a maternal environment for up to 9.5 months,” Please add references to support this statement.

5. “These demographic factors are important to consider because current evidence suggests those from lower SES and/or first-time mothers may be at increased risk of developing perinatal depression; however, there is conflicting evidence for education being a risk factor versus a protective factor.” Please add the corresponding references. In addition, the authors did not discuss the reasons for the contradictory results, please explain it.

6. “Conversely, it has been indicated that perinatal depression may result in early cessation or that difficulties with breastfeeding may contribute to perinatal depression symptoms.” Please add the corresponding references.

7. “The bioavailability of essential amino acids (e.g., tryptophan, competitor amino acids), the precursors to a number of neurotransmitters commonly associated with psychiatric conditions, depends on dietary intake.” Please add references to support this statement.

8. “Conversely, it has been suggested that TRP has a higher affinity for the BBB than for albumin, and albumin bound TRP close to the BBB may separate from albumin to then transport across the BBB.” Please add references to support this statement.

9. The authors stated that “Moreover, IL-1β was found to be negatively associated with depression scores across four-time points (i.e., three trimesters, one postpartum time point).” However, the authors also stated that “Plasma IL-10 (anti-inflammatory cytokine) and IL-1β (pro-inflammatory cytokine) were not associated with increased risk for PPD.” These two findings may seem contradictory. Please explain it in a way that is easy for the readers to understand and add the corresponding references.

10. “While sources of oxidative stress vary, evidence suggests the sources are largely related to environmental and lifestyle factors.” Please add references to support this statement.

11. “Additionally, though not a variable noted in any of the included studies, nausea and vomiting due to“morning sickness” or hyperemesis gravidarum (severe type of morning sickness) occurs in roughly 70% of pregnancies.” Please add references to support this data.

12. “The findings related to substance abuse, history of suicide attempts or current suicidal ideation, and exercise are consistent with existing literature” Please add the corresponding references.

13. The authors stated that “Interestingly, non-perinatal specific research that began examining the impact of COVID-19 on food insecurity found food insecurity to disproportionately impact racial and ethnic groups, and the states with the highest projected food insecurity rates based on overall population occurred in states that also have some of the highest maternal mortality rates (i.e., Louisiana, Texas)” However, this example does not seem to support the conclusion that food insecurity is a potential predictor of perinatal depression. Because in this example, the relationship between food insecurity and perinatal depression appears to be unclear. In addition, Please check if Ref. [76] is correctly cited.

14. “Some evidence suggests the disruption of the serotonergic system is more prominent in biologically born females compared to males, and that the dysregulation of serotonin may partly explain why biologically born females experience depression at two times the rate of biologically born males.” Please add the corresponding references.

Reviewer #2: The present review aimed to focus on the biopsychosocial and behavioral determinants of perinatal depression using an integrative approach.

In the results, they put a lot of work on the Funding sources and the region (US and non-US based research), which, in my opinion, was not so valuable for the objectives of the review. This part may be shortened to make the review concentrate on the objectives.

The participant characteristics could be included in the part of the social determinants.

And there are two other important things should be included/ discussed:

1、When the depression is determined is very important, since the emotion of the pregnant / postpartum women can vary much in the process.

2、Another important thing, is the health of the fetus/ offspring, or pregnancy outcome, maternal emotion can be significantly affected by these.

6. PLOS authors have the option to publish the peer review history of their article (what does this mean?). If published, this will include your full peer review and any attached files.

Reviewer #1: No

Reviewer #2: No

---

## [Author Response · Author response to Decision Letter 0]

11 Oct 2023

Thank you for your time and thoughtful feedback. We have revised our manuscript based on this feedback and addressed each of your comments in the response to reviewers table.

---

## [Decision Letter · Decision Letter 1]

1 Jul 2024

A sum of its parts: A systematic review evaluating biopsychosocial and behavioral determinants of perinatal depression

PONE-D-23-17835R1

Dear Dr. Longoria,

We’re pleased to inform you that your manuscript has been judged scientifically suitable for publication and will be formally accepted for publication once it meets all outstanding technical requirements.

Kind regards,

Rita Amiel Castro

Academic Editor

PLOS ONE

Additional Editor Comments (optional):

Reviewers' comments:

Reviewer's Responses to Questions

**Comments to the Author**

1. If the authors have adequately addressed your comments raised in a previous round of review and you feel that this manuscript is now acceptable for publication, you may indicate that here to bypass the “Comments to the Author” section, enter your conflict of interest statement in the “Confidential to Editor” section, and submit your "Accept" recommendation.

Reviewer #3: All comments have been addressed

2. Is the manuscript technically sound, and do the data support the conclusions?

Reviewer #3: Yes

3. Has the statistical analysis been performed appropriately and rigorously? 

Reviewer #3: Yes

4. Have the authors made all data underlying the findings in their manuscript fully available?

Reviewer #3: Yes

5. Is the manuscript presented in an intelligible fashion and written in standard English?

Reviewer #3: Yes

6. Review Comments to the Author

Reviewer #3: In my view the manuscript has been improved accordingly. I have reviewed the document and I believe it can be published.

7. PLOS authors have the option to publish the peer review history of their article (what does this mean?). If published, this will include your full peer review and any attached files.

Reviewer #3: No

---

## [Editor Report · Acceptance letter]

4 Jul 2024

PONE-D-23-17835R1 

PLOS ONE

Dear Dr. Longoria, 

I'm pleased to inform you that your manuscript has been deemed suitable for publication in PLOS ONE. Congratulations! Your manuscript is now being handed over to our production team.

Kind regards, 

on behalf of

Dr. Rita Amiel Castro 

Academic Editor

PLOS ONE